# Loss of plasticity in deep continual learning

Shibhansh Dohare[1✉], J. Fernando Hernandez-Garcia[1], Qingfeng Lan[1], Parash Rahman[1], A. Rupam Mahmood[1,2] & Richard S. Sutton[1,2]

Artificial neural networks, deep-learning methods and the backpropagation algorithm[1] form the foundation of modern machine learning and artificial intelligence. These methods are almost always used in two phases, one in which the weights of the network are updated and one in which the weights are held constant while the network is used or evaluated. This contrasts with natural learning and many applications, which require continual learning. It has been unclear whether or not deep learning methods work in continual learning settings. Here we show that they do not—that standard deep-learning methods gradually lose plasticity in continual-learning settings until they learn no better than a shallow network. We show such loss of plasticity using the classic ImageNet dataset and reinforcement-learning problems across a wide range of variations in the network and the learning algorithm. Plasticity is maintained indefinitely only by algorithms that continually inject diversity into the network, such as our continual backpropagation algorithm, a variation of backpropagation in which a small fraction of less-used units are continually and randomly reinitialized. Our results indicate that methods based on gradient descent are not enough—that sustained deep learning requires a random, non-gradient component to maintain variability and plasticity.

Machine learning and artificial intelligence have made remarkable progress in the past decade, with landmark successes in natural-language processing[2,3], biology[4], game playing[5–8] and robotics[9,10]. All these systems use artificial neural networks, whose computations are inspired by the operation of human and animal brains. Learning in these networks refers to computational algorithms for changing the strengths of their connection weights (computational synapses). The most important modern learning methods are based on stochastic gradient descent (SGD) and the backpropagation algorithm, ideas that originated at least four decades ago but are much more powerful today because of the availability of vastly greater computer power. The successes are also because of refinements of the learning and training techniques that together make the early ideas effective in much larger and more deeply layered networks. These methodologies are collectively referred to as deep learning.

Despite its successes, deep learning has difficulty adapting to changing data. Because of this, in almost all applications, deep learning is restricted to a special training phase and then turned off when the network is actually used. For example, large language models such as ChatGPT are trained on a large generic training set and then fine-tuned on smaller datasets specific to an application or to meet policy and safety goals, but finally their weights are frozen before the network is released for use. With current methods, it is usually not effective to simply continue training on new data when they become available. The effect of the new data is either too large or too small and not properly balanced with the old data. The reasons for this are not well understood and there is not yet a clear solution. In practice, the most common strategy for incorporating substantial new data has been simply to discard the old network and train a new one from scratch on the old and new data together[11,12]. When the network is a large language model and the data are a substantial portion of the internet, then each retraining may cost millions of dollars in computation. Moreover, a wide range of real-world applications require adapting to change. Change is ubiquitous in learning to anticipate markets and human preferences and in gaming, logistics and control systems. Deep-learning systems would be much more powerful if they, like natural-learning systems, were capable of continual learning.

Here we show systematically that standard deep-learning methods lose their ability to learn with extended training on new data, a phenomenon that we call loss of plasticity. We use classic datasets, such as ImageNet and CIFAR-100, modified for continual learning, and standard feed-forward and residual networks with a wide variety of standard learning algorithms. Loss of plasticity in artificial neural networks was first shown at the turn of the century in the psychology literature[13–15], before the development of deep-learning methods. Plasticity loss with modern methods was visible in some recent works[11,16–18] and most recently has begun to be explored explicitly[12,19–27]. Loss of plasticity is different from catastrophic forgetting, which concerns poor performance on old examples even if they are not presented again[28–30].

Although standard deep-learning methods lose plasticity with extended learning, we show that a simple change enables them to maintain plasticity indefinitely in both supervised and reinforcement learning. Our new algorithm, continual backpropagation, is exactly like classical backpropagation except that a tiny proportion of less-used units are reinitialized on each step much as they were all initialized at the start of training. Continual backpropagation is inspired by a long history of methods for automatically generating and testing features, starting with Selfridge's Pandemonium in 1959

[1]Department of Computing Science, University of Alberta, Edmonton, Alberta, Canada. [2]Canada CIFAR AI Chair, Alberta Machine Intelligence Institute (Amii), Edmonton, Alberta, Canada. ✉e-mail: dohare@ualberta.ca

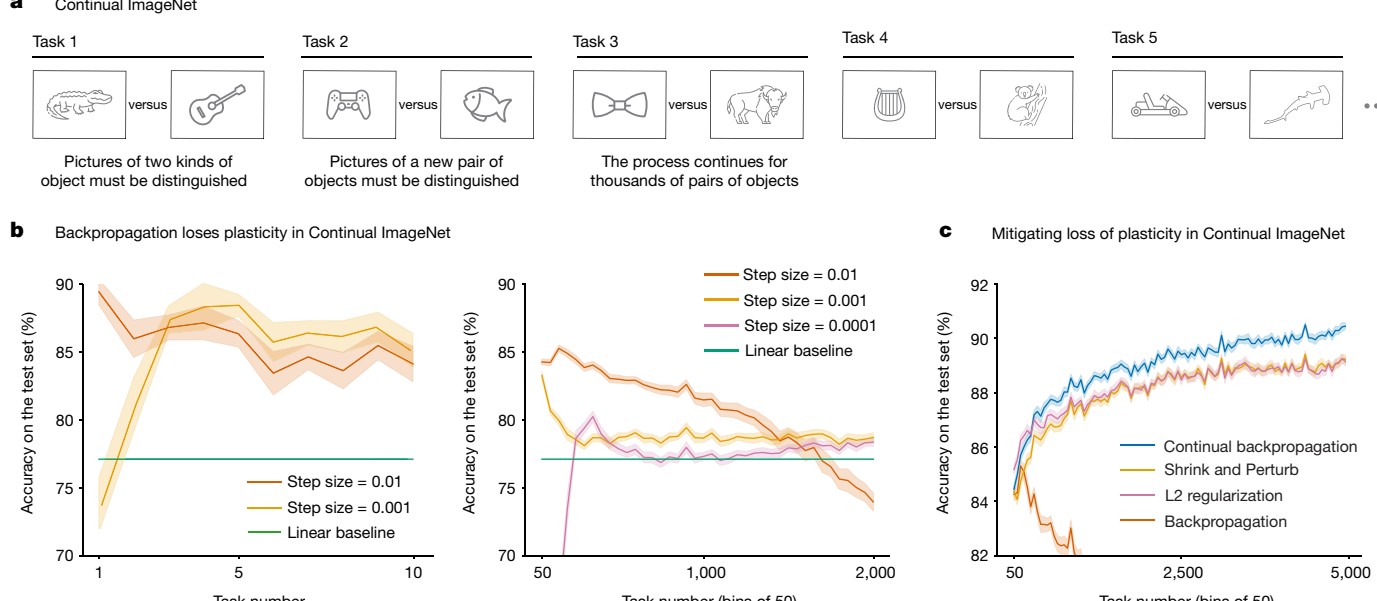

**a** Continual ImageNet

Task 1 Task 2 Task 3 Task 4 Task 5

Pictures of two kinds of object must be distinguished Pictures of a new pair of objects must be distinguished The process continues for thousands of pairs of objects

**b** Backpropagation loses plasticity in Continual ImageNet

**c** Mitigating loss of plasticity in Continual ImageNet

**Fig. 1 | Plasticity loss in Continual ImageNet. a–c**, In a sequence of binary classification tasks using ImageNet pictures (**a**), the conventional backpropagation algorithm loses plasticity at all step sizes (**b**), whereas the continual backpropagation, L2 regularization and Shrink and Perturb algorithms maintain plasticity, apparently indefinitely (**c**). All results are averaged over 30 runs; the solid lines represent the mean and the shaded regions correspond to ±1 standard error.

(refs. 19,20,31–35). The effectiveness of continual backpropagation shows that the problem of plasticity loss is not inherent in artificial neural networks.

## Plasticity loss in supervised learning

The primary purpose of this article is to demonstrate loss of plasticity in standard deep-learning systems. For the demonstration to be convincing, it must be systematic and extensive. It must consider a wide range of standard deep-learning networks, learning algorithms and parameter settings. For each of these, the experiments must be run long enough to expose long-term plasticity loss and be repeated enough times to obtain statistically significant results. Altogether, more computation is needed by three or four orders of magnitude compared with what would be needed to train a single network. For example, a systematic study with large language models would not be possible today because just a single training run with one of these networks would require computation costing millions of dollars. Fortunately, advances in computer hardware have continued apace since the development of deep learning and systematic studies have become possible with the deep-learning networks used earlier and with some of the longer-lived test problems. Here we use ImageNet, a classic object-recognition test bed[36], which played a pivotal role in the rise of deep learning[37] and is still influential today.

The ImageNet database comprises millions of images labelled by nouns (classes) such as types of animal and everyday object. The typical ImageNet task is to guess the label given an image. The standard way to use this dataset is to partition it into training and test sets. A learning system is first trained on a set of images and their labels, then training is stopped and performance is measured on a separate set of test images from the same classes. To adapt ImageNet to continual learning while minimizing all other changes, we constructed a sequence of binary classification tasks by taking the classes in pairs. For example, the first task might be to distinguish cats from houses and the second might be to distinguish stop signs from school buses. With the 1,000 classes in our dataset, we were able to form half a million binary classification tasks in this way. For each task, a deep-learning network was first trained on a subset of the images for the two classes and then its performance was measured on a separate test set for the classes. After training and testing on one task, the next task began with a different pair of classes. We call this problem 'Continual ImageNet'. In Continual ImageNet, the difficulty of tasks remains the same over time. A drop in performance would mean the network is losing its learning ability, a direct demonstration of loss of plasticity.

We applied a wide variety of standard deep-learning networks to Continual ImageNet and tested many learning algorithms and parameter settings. To assess the performance of the network on a task, we measured the percentage of test images that were correctly classified. The results shown in Fig. 1b are representative; they are for a feed-forward convolutional network and for a training procedure, using unmodified backpropagation, that performed well on this problem in the first few tasks.

Although these networks learned up to 88% correct on the test set of the early tasks (Fig. 1b, left panel), by the 2,000th task, they had lost substantial plasticity for all values of the step-size parameter (right panel). Some step sizes performed well on the first two tasks but then much worse on subsequent tasks, eventually reaching a performance level below that of a linear network. For other step sizes, performance rose initially and then fell and was only slightly better than the linear network after 2,000 tasks. We found this to be a common pattern in our experiments: for a well-tuned network, performance first improves and then falls substantially, ending near or below the linear baseline. We have observed this pattern for many network architectures, parameter choices and optimizers. The specific choice of network architecture, algorithm parameters and optimizers affected when the performance started to drop, but a severe performance drop occurred for a wide range of choices. The failure of standard deep-learning methods to learn better than a linear network in later tasks is direct evidence that these methods do not work well in continual-learning problems.

Algorithms that explicitly keep the weights of the network small were an exception to the pattern of failure and were often able to maintain plasticity and even improve their performance over many tasks, as shown in Fig. 1c. L2 regularization adds a penalty for large weights;

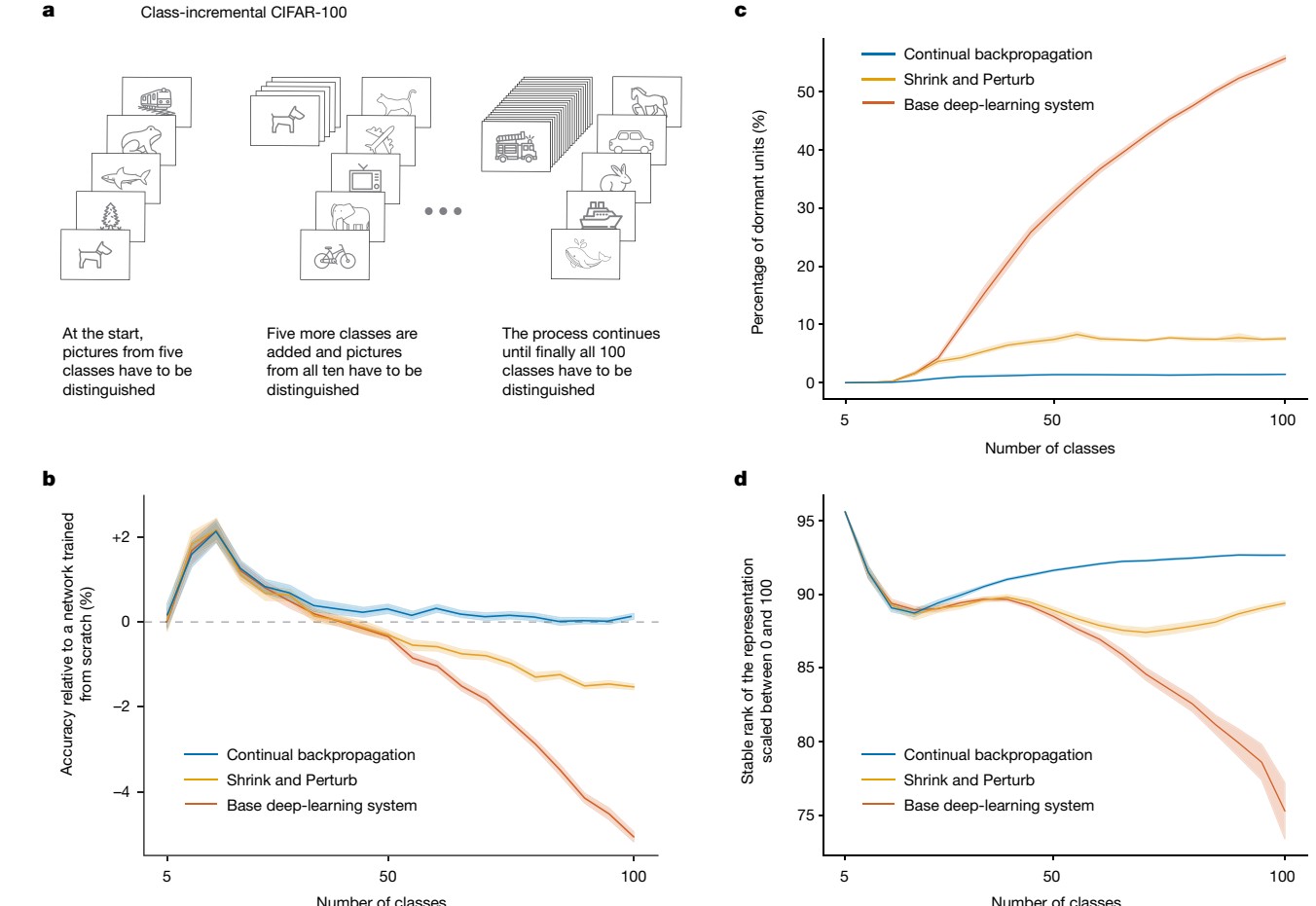

**Fig. 2 | Plasticity loss in class-incremental CIFAR-100. a**, An incrementally growing image-classification problem. **b**, Initially, accuracy is improved by incremental training compared with a network trained from scratch, but after 40 classes, accuracy degrades substantially in a base deep-learning system, less so for a Shrink and Perturb learning system and not at all for a learning system based on continual backpropagation. **c**, The number of network units that are active less than 1% of the time increases rapidly for the base deep-learning system, but less so for Shrink and Perturb and continual backpropagation systems. **d**, A low stable rank means that the units of a network do not provide much diversity; the base deep-learning system loses much more diversity than the Shrink and Perturb and continual backpropagation systems. All results are averaged over 30 runs; the solid lines represent the mean and the shaded regions correspond to ±1 standard error.

augmenting backpropagation with this enabled the network to continue improving its learning performance over at least 5,000 tasks. The Shrink and Perturb algorithm[11], which includes L2 regularization, also performed well. Best of all was our continual backpropagation algorithm, which we discuss later. For all algorithms, we tested a wide range of parameter settings and performed many independent runs for statistical significance. The presented curves are the best representative of each algorithm.

For a second demonstration, we chose to use residual networks, class-incremental continual learning and the CIFAR-100 dataset. Residual networks include layer-skipping connections as well as the usual layer-to-layer connections of conventional convolutional networks. The residual networks of today are more widely used and produce better results than strictly layered networks[38]. Class-incremental continual learning[39] involves sequentially adding new classes while testing on all classes seen so far. In our demonstration, we started with training on five classes and then successively added more, five at a time, until all 100 were available. After each addition, the networks were trained and performance was measured on all available classes. We continued training on the old classes (unlike in most work in class-incremental learning) to focus on plasticity rather than on forgetting.

In this demonstration, we used an 18-layer residual network with a variable number of heads, adding heads as new classes were added.

We also used further deep-learning techniques, including batch normalization, data augmentation, L2 regularization and learning-rate scheduling. These techniques are standardly used with residual networks and are necessary for good performance. We call this our base deep-learning system.

As more classes are added, correctly classifying images becomes more difficult and classification accuracy would decrease even if the network maintained its ability to learn. To factor out this effect, we compare the accuracy of our incrementally trained networks with networks that were retrained from scratch on the same subset of classes. For example, the network that was trained first on five classes, and then on all ten classes, is compared with a network retrained from scratch on all ten classes. If the incrementally trained network performs better than a network retrained from scratch, then there is a benefit owing to training on previous classes, and if it performs worse, then there is genuine loss of plasticity.

The red line in Fig. 2b shows that incremental training was initially better than retraining, but after 40 classes, the incrementally trained network showed loss of plasticity that became increasingly severe. By the end, when all 100 classes were available, the accuracy of the incrementally trained base system was 5% lower than the retrained network (a performance drop equivalent to that of removing a notable algorithmic advance, such as batch normalization). Loss of plasticity was less

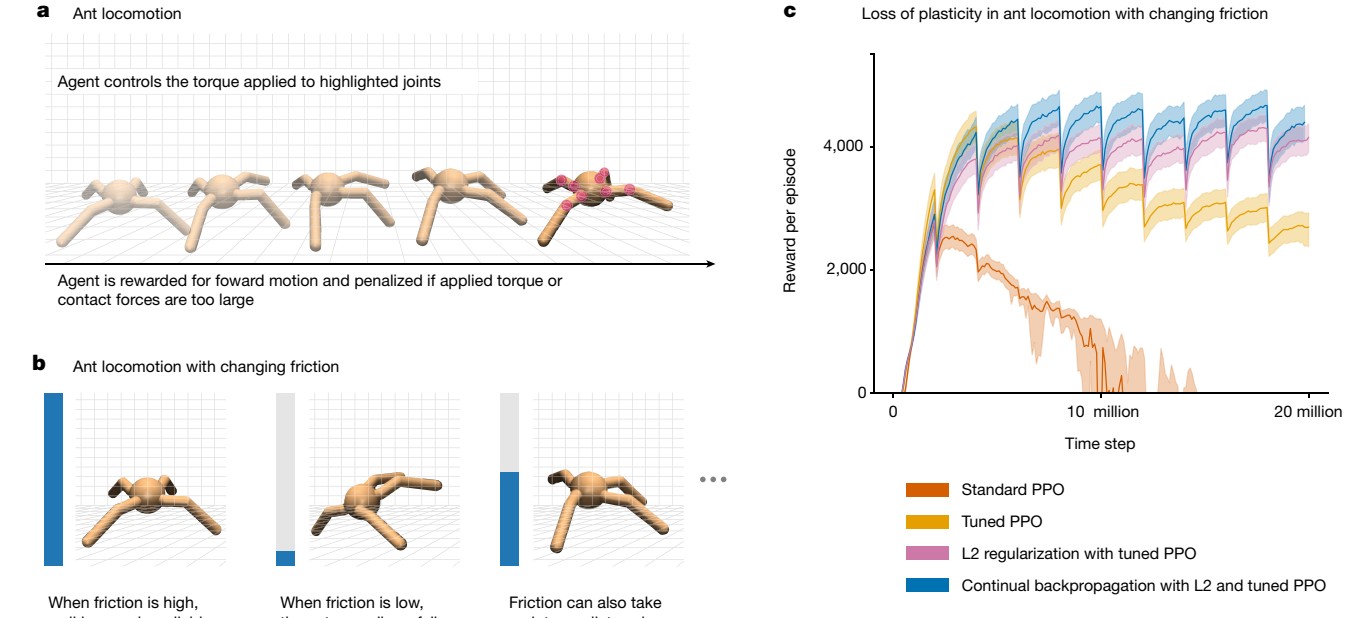

**a** Ant locomotion

Agent controls the torque applied to highlighted joints

Agent is rewarded for foward motion and penalized if applied torque or contact forces are too large

**b** Ant locomotion with changing friction

When friction is high, walking can be reliable

When friction is low, the ant may slip or fall

Friction can also take on intermediate values

**c** Loss of plasticity in ant locomotion with changing friction

— Standard PPO
— Tuned PPO
— L2 regularization with tuned PPO
— Continual backpropagation with L2 and tuned PPO

**Fig. 3 | Maintaining plasticity in a non-stationary reinforcement-learning problem. a**, The reinforcement-learning agent controls torques at the eight joints of the simulated ant (red circles) to maximize forward motion and minimize penalties. **b**, Here we use a version of the ant problem in which the friction on contact with the ground is abruptly changed every 2 million time steps. **c**, The standard PPO learning algorithm fails catastrophically on the non-stationary ant problem. If the optimizer of PPO (Adam) is tuned in a custom way, then the failure is less severe, but adding continual backpropagation or L2 regularization is necessary to perform well indefinitely. These results are averaged over 100 runs; the solid lines represent the mean and the shaded regions represent the 95% bootstrapped confidence interval.

severe when Shrink and Perturb was added to the learning algorithm (in the incrementally trained network) and was eliminated altogether when continual backpropagation (see the 'Maintaining plasticity through variability and selective preservation' section) was added. These additions also prevented units of the network from becoming inactive or redundant, as shown in Fig. 2c,d.

This demonstration involved larger networks and required more computation, but still we were able to perform extensive systematic tests. We found a robust pattern in the results that was similar to what we found in ImageNet. In both cases, deep-learning networks exhibited substantial loss of plasticity. Altogether, these results, along with other extensive results in Methods, constitute substantial evidence of plasticity loss.

## Plasticity loss in reinforcement learning

Continual learning is essential to reinforcement learning in ways that go beyond its importance in supervised learning. Not only can the environment change but the behaviour of the learning agent can also change, thereby influencing the data it receives even if the environment is stationary. For this reason, the need for continual learning is often more apparent in reinforcement learning, and reinforcement learning is an important setting in which to demonstrate the tendency of deep learning towards loss of plasticity.

Nevertheless, it is challenging to demonstrate plasticity loss in reinforcement learning in a systematic and rigorous way. In part, this is because of the great variety of algorithms and experimental settings that are commonly used in reinforcement-learning research. Algorithms may learn value functions, behaviours or both simultaneously and may involve replay buffers, world models and learned latent states. Experiments may be episodic, continuing or offline. All of these choices involve several embedded choices of parameters. More fundamentally, reinforcement-learning algorithms affect the data seen by the agent. The learning ability of an algorithm is thus confounded with its ability to generate informative data. Finally, and in

part because of the preceding, reinforcement-learning results tend to be more stochastic and more widely varying than in supervised learning. Altogether, demonstration of reinforcement-learning abilities, particularly negative results, tends to require more runs and generally much more experimental work and thus inevitably cannot be as definitive as in supervised learning.

Our first demonstration involves a reinforcement-learning algorithm applied to a simulated ant-like robot tasked with moving forwards as rapidly and efficiently as possible. The agent–environment interaction comprises a series of episodes, each beginning in a standard state and lasting up to 1,000 time steps. On each time step, the agent receives a reward depending on the forward distance travelled and the magnitude of its action (see Methods for details). An episode terminates in fewer than 1,000 steps if the ant jumps too high instead of moving forwards, as often happens early in learning. In the results to follow, we use the cumulative reward during an episode as our primary performance measure. To make the task non-stationary (and thereby emphasize plasticity), the coefficient of friction between the feet of the ant and the floor is changed after every 2 million time steps (but only at an episode boundary; details in Methods). For fastest walking, the agent must adapt (relearn) its way of walking each time the friction changes. For this experiment, we used the proximal policy optimization (PPO) algorithm[40]. PPO is a standard deep reinforcement-learning algorithm based on backpropagation. It is widely used, for example, in robotics[9], in playing real-time strategy games[41] and in aligning large language models from human feedback[42].

PPO performed well (see the red line in Fig. 3c) for the first 2 million steps, up until the first change in friction, but then performed worse and worse. Note how the performance of the other algorithms in Fig. 3c decreased each time the friction changed and then recovered as the agent adapted to the new friction, giving the plot a sawtooth appearance. PPO augmented with a specially tuned Adam optimizer[24,43] performed much better (orange line in Fig. 3c) but still performed much worse over successive changes after the first two, indicating substantial loss of plasticity. On the other hand, PPO augmented with

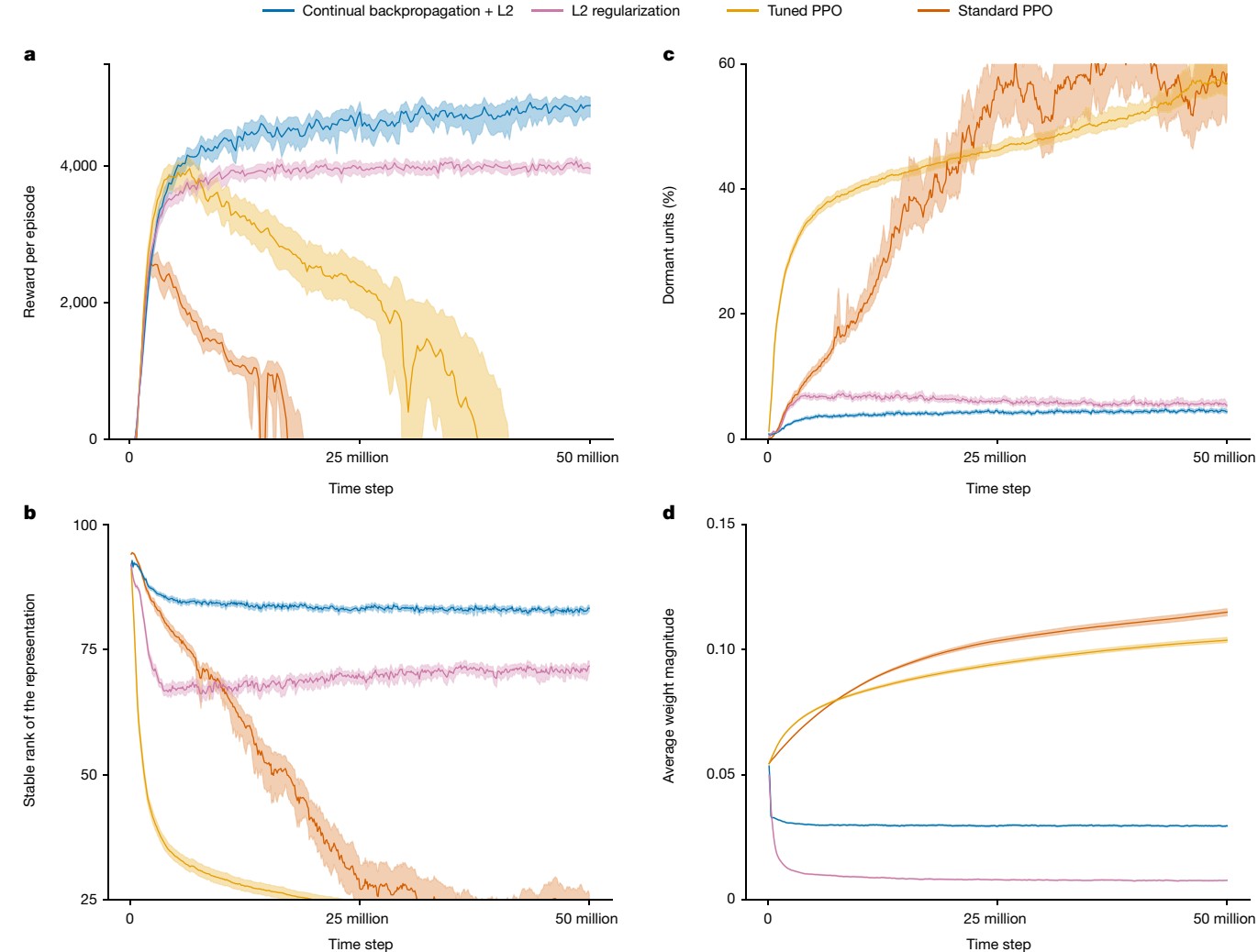

**Fig. 4 | Results on a stationary ant-locomotion problem. a,** The four reinforcement-learning algorithms performed similarly on this and the non-stationary problem (compare with Fig. 3c). **b,c,** A closer look inside the networks reveals a similar pattern as in supervised learning (compare with Fig. 2c,d). **d,** The absolute values of the weights of the networks increased steadily under standard and tuned PPO, whereas they decreased and stayed small under L2 regularization with or without continual backpropagation. These results are averaged over 30 runs; the solid lines represent the mean and the shaded regions represent the 95% bootstrapped confidence interval.

L2 regularization and continual backpropagation largely maintained their plasticity as the problem changed.

Now consider the same ant-locomotion task except with the coefficient of friction held constant at an intermediate value over 50 million time steps. The red line in Fig. 4a shows that the average performance of PPO increased for about 3 million steps but then collapsed. After 20 million steps, the ant is failing every episode and is unable to learn to move forwards efficiently. The red lines in the other panels of Fig. 4 provide further insight into the loss of plasticity of PPO. They suggest that the network may be losing plasticity in the same way as in our supervised learning results (see Fig. 2 and Extended Data Fig. 3c). In both cases, most of the network's units became dormant during the experiment, and the network markedly lost stable rank. The addition of L2 regularization mitigated the performance degradation by preventing continual growth of weights but also resulted in very small weights (Fig. 4d), which prevented the agent from committing to good behaviour. The addition of continual backpropagation performed better overall. We present results for continual backpropagation only with (slight) L2 regularization, because without it, performance was highly sensitive to parameter settings. These results show that plasticity loss can be catastrophic in both deep reinforcement learning as well as deep supervised learning.

## Maintaining plasticity

Surprisingly, popular methods such as Adam, Dropout and normalization actually increased loss of plasticity (see Extended Data Fig. 4a). L2 regularization, on the other hand, reduced loss of plasticity in many cases (purple line in Figs. 1, 3 and 4). L2 regularization stops the weights from becoming too large by moving them towards zero at each step. The small weights allow the network to remain plastic. Another existing method that reduced loss of plasticity is Shrink and Perturb[11] (orange line in Figs. 1 and 2). Shrink and Perturb is L2 regularization plus small random changes in weights at each step. The injection of variability into the network can reduce dormancy and increase the diversity of the representation (Figs. 2 and 4). Our results indicate that non-growing weights and sustained variability in the network may be important for maintaining plasticity.

We now describe a variation of the backpropagation algorithm that is explicitly designed to inject variability into the network and keep some of its weights small. Conventional backpropagation has two main parts: initialization with small random weights before training and then gradient descent at each training step. The initialization provides variability initially, but, as we have seen, with continued training, variability tends to be lost, as well as plasticity along with it. To maintain

the variability, our new algorithm, continual backpropagation, reinitializes a small number of units during training, typically fewer than one per step. To prevent disruption of what the network has already learned, only the least-used units are considered for reinitialization. See Methods for details.

The blue line in Fig. 1c shows the performance of continual backpropagation on Continual ImageNet. It mitigated loss of plasticity in Continual ImageNet while outperforming existing methods. Similarly, the blue lines in Fig. 2 show the performance of continual backpropagation on class-incremental CIFAR-100 and its effect on the evolution of dormant units and stable rank. Continual backpropagation fully overcame loss of plasticity, with a high stable rank and almost no dead units throughout learning.

In reinforcement learning, continual backpropagation was applied together with L2 regularization (a small amount of regularization was added to prevent excessive sensitivity to parameters in reinforcement-learning experiments). The blue line in Fig. 3 shows the performance of PPO with continual backpropagation on the ant-locomotion problem with changing friction. PPO with continual backpropagation performed much better than standard PPO, with little or no loss of plasticity. On the ant-locomotion problem with constant friction (Fig. 4), PPO with continual backpropagation continued improving throughout the experiment. The blue lines in Fig. 4b–d show the evolution of the correlates of loss of plasticity when we used continual backpropagation. PPO with continual backpropagation had few dormant units, a high stable rank and an almost constant average weight magnitude.

Our results are consistent with the idea that small weights reduce loss of plasticity and that a continual injection of variability further mitigates loss of plasticity. Although Shrink and Perturb adds variability to all weights, continual backpropagation does so selectively, and this seems to enable it to better maintain plasticity. Continual backpropagation involves a form of variation and selection in the space of neuron-like units, combined with continuing gradient descent. The variation and selection is reminiscent of trial-and-error processes in evolution and behaviour[44–47] and has precursors in many earlier ideas, including Keifer–Wolfowitz methods[48] and restart methods[49] in engineering and feature-search methods in machine learning[31–35,50]. Continual backpropagation brings a form of this old idea to modern deep learning. However, it is just one variation of this idea; other variations are possible and some of these have been explored in recent work[25,27]. We look forward to future work that explicitly compares and further refines these variations.

## Discussion

Deep learning is an effective and valuable technology in settings in which learning occurs in a special training phase and not thereafter. In settings in which learning must continue, however, we have shown that deep learning does not work. By deep learning, we mean the existing standard algorithms for learning in multilayer artificial neural networks and by not work, we mean that, over time, they fail to learn appreciably better than shallow networks. We have shown such loss of plasticity using supervised-learning datasets and reinforcement-learning tasks on which deep learning has previously excelled and for a wide range of networks and standard learning algorithms. Taking a closer look, we found that, during training, many of the networks' neuron-like units become dormant, overcommitted and similar to each other, hampering the ability of the networks to learn new things. As they learn, standard deep-learning networks gradually and irreversibly lose their diversity and thus their ability to continue learning. Plasticity loss is often severe when learning continues for many tasks, but may not occur at all for small numbers of tasks.

The problem of plasticity loss is not intrinsic to deep learning. Deep artificial neural networks trained by gradient descent are perfectly capable of maintaining their plasticity, apparently indefinitely, as we have shown with the Shrink and Perturb algorithm and particularly with the new continual backpropagation algorithm. Both of these algorithms extend standard deep learning by adding a source of continuing variability to the weights of the network, and continual backpropagation restricts this variability to the units of the network that are at present least used, minimizing damage to the operation of the network. That is, continual backpropagation involves a form of variation and selection in the space of neuron-like units, combined with continuing gradient descent. This idea has many historical antecedents and will probably require further development to reach its most effective form.

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

## Methods

### Specifics of continual backpropagation

Continual backpropagation selectively reinitializes low-utility units in the network. Our utility measure, called the contribution utility, is defined for each connection or weight and each unit. The basic intuition behind the contribution utility is that the magnitude of the product of units' activation and outgoing weight gives information about how valuable this connection is to its consumers. If the contribution of a hidden unit to its consumer is small, its contribution can be overwhelmed by contributions from other hidden units. In such a case, the hidden unit is not useful to its consumer. We define the contribution utility of a hidden unit as the sum of the utilities of all its outgoing connections. The contribution utility is measured as a running average of instantaneous contributions with a decay rate, $\eta$, which is set to 0.99 in all experiments. In a feed-forward neural network, the contribution utility, $\mathbf{u}_l[i]$, of the $i$th hidden unit in layer $l$ at time $t$ is updated as

$$\mathbf{u}_l[i] = \eta \times \mathbf{u}_l[i] + (1 - \eta) \times |\mathbf{h}_{l,i,t}| \times \sum_{k=1}^{n_{l+1}} |\mathbf{w}_{l,i,k,t}|, \qquad (1)$$

in which $\mathbf{h}_{l,i,t}$ is the output of the $i$th hidden unit in layer $l$ at time $t$, $\mathbf{w}_{l,i,k,t}$ is the weight connecting the $i$th unit in layer $l$ to the $k$th unit in layer $l+1$ at time $t$ and $n_{l+1}$ is the number of units in layer $l+1$.

When a hidden unit is reinitialized, its outgoing weights are initialized to zero. Initializing the outgoing weights as zero ensures that the newly added hidden units do not affect the already learned function. However, initializing the outgoing weight to zero makes the new unit vulnerable to immediate reinitialization, as it has zero utility. To protect new units from immediate reinitialization, they are protected from a reinitialization for maturity threshold $m$ number of updates. We call a unit mature if its age is more than $m$. Every step, a fraction of mature units $\rho$, called the replacement rate, is reinitialized in every layer.

The replacement rate $\rho$ is typically set to a very small value, meaning that only one unit is replaced after hundreds of updates. For example, in class-incremental CIFAR-100 (Fig. 2) we used continual backpropagation with a replacement rate of $10^{-5}$. The last layer of the network in that problem had 512 units. At each step, roughly $512 \times 10^{-5} = 0.00512$ units are replaced. This corresponds to roughly one replacement after every $1/0.00512 \approx 200$ updates or one replacement after every eight epochs on the first five classes.

The final algorithm combines conventional backpropagation with selective reinitialization to continually inject random units from the initial distribution. Continual backpropagation performs a gradient descent and selective reinitialization step at each update. Algorithm 1 specifies continual backpropagation for a feed-forward neural network. In cases in which the learning system uses mini-batches, the instantaneous contribution utility can be used by averaging the utility over the mini-batch instead of keeping a running average to save computation (see Extended Data Fig. 5d for an example). Continual backpropagation overcomes the limitation of previous work[34,35] on selective reinitialization and makes it compatible with modern deep learning.

### Algorithm 1. Continual backpropagation for a feed-forward network with $L$ layers

Set replacement rate $\rho$, decay rate $\eta$ and maturity threshold $m$
Initialize the weights $\mathbf{w}_0, \ldots, \mathbf{w}_{L-1}$, in which $\mathbf{w}_l$ is sampled from distribution $d_l$
Initialize utilities $\mathbf{u}_1, \ldots, \mathbf{u}_{L-1}$, number of units to replace $c_1, \ldots, c_{L-1}$, and ages $\mathbf{a}_1, \ldots, \mathbf{a}_{L-1}$ to 0
**For** each input $\mathbf{x}_t$ **do**
Forward pass: pass $\mathbf{x}_t$ through the network to get the prediction $\widehat{\mathbf{y}}_t$
Evaluate: receive loss $l(\mathbf{x}_t, \widehat{\mathbf{y}}_t)$
Backward pass: update the weights using SGD or one of its variants
**For** layer $l$ in $1: L - 1$ **do**

Update age: $\mathbf{a}_l = \mathbf{a}_l + 1$
Update unit utility: see equation (1)
Find eligible units: $n_{\text{eligible}}$ = number of units with age greater than $m$
Update number of units to replace: $c_l = c_l + n_{\text{eligible}} \times \rho$
**If** $c_l > 1$
Find the unit with smallest utility and record its index as $r$
Reinitialize input weights: resample $\mathbf{w}_{l-1}[:, r]$ from distribution $d_l$
Reinitialize output weights: set $\mathbf{w}_l[r, :]$ to 0
Reinitialize utility and age: set $\mathbf{u}_l[r] = 0$ and $\mathbf{a}_l[r] = 0$
Update number of units to replace: $c_l = c_l - 1$
**End For**
**End For**

### Details of Continual ImageNet

The ImageNet database we used consists of 1,000 classes, each of 700 images. The 700 images for each class were divided into 600 images for a training set and 100 images for a test set. On each binary classification task, the deep-learning network was first trained on the training set of 1,200 images and then its classification accuracy was measured on the test set of 200 images. The training consisted of several passes through the training set, called epochs. For each task, all learning algorithms performed 250 passes through the training set using mini-batches of size 100. All tasks used the downsampled $32 \times 32$ version of the ImageNet dataset, as is often done to save computation[51].

All algorithms on Continual ImageNet used a convolutional network. The network had three convolutional-plus-max-pooling layers, followed by three fully connected layers, as detailed in Extended Data Table 3. The final layer consisted of just two units, the heads, corresponding to the two classes. At task changes, the input weights of the heads were reset to zero. Resetting the heads in this way can be viewed as introducing new heads for the new tasks. This resetting of the output weights is not ideal for studying plasticity, as the learning system gets access to privileged information on the timing of task changes (and we do not use it in other experiments in this paper). We use it here because it is the standard practice in deep continual learning for this type of problem in which the learning system has to learn a sequence of independent tasks[52].

In this problem, we reset the head of the network at the beginning of each task. It means that, for a linear network, the whole network is reset. That is why the performance of a linear network will not degrade in Continual ImageNet. As the linear network is a baseline, having a low-variance estimate of its performance is desirable. The value of this baseline is obtained by averaging over thousands of tasks. This averaging gives us a much better estimate of its performance than other networks.

The network was trained using SGD with momentum on the cross-entropy loss and initialized once before the first task. The momentum hyperparameter was 0.9. We tested various step-size parameters for backpropagation but only presented the performance for step sizes 0.01, 0.001 and 0.0001 for clarity of Fig. 1b. We performed 30 runs for each hyperparameter value, varying the sequence of tasks and other randomness. Across different hyperparameters and algorithms, the same sequences of pairs of classes were used.

We now describe the hyperparameter selection for L2 regularization, Shrink and Perturb and continual backpropagation. The main text presents the results for these algorithms on Continual ImageNet in Fig. 1c. We performed a grid search for all algorithms to find the set of hyperparameters that had the highest average classification accuracy over 5,000 tasks. The values of hyperparameters used for the grid search are described in Extended Data Table 2. L2 regularization has two hyperparameters, step size and weight decay. Shrink and Perturb has three hyperparameters, step size, weight decay and noise variance. We swept over two hyperparameters of continual backpropagation: step size and replacement rate. The maturity threshold in continual backpropagation was set to 100. For both backpropagation and L2

regularization, the performance was poor for step sizes of 0.1 or 0.003. We chose to only use step sizes of 0.03 and 0.01 for continual backpropagation and Shrink and Perturb. We performed ten independent runs for all sets of hyperparameters. Then we performed another 20 runs to complete 30 runs for the best-performing set of hyperparameters to produce the results in Fig. 1c.

## Class-incremental CIFAR-100

In the class-incremental CIFAR-100, the learning system gets access to more and more classes over time. Classes are provided to the learning system in increments of five. First, it has access to just five classes, then ten and so on, until it gets access to all 100 classes. The learning system is evaluated on the basis of how well it can discriminate between all the available classes at present. The dataset consists of 100 classes with 600 images each. The 600 images for each class were divided into 450 images to create a training set, 50 for a validation set and 100 for a test set. Note that the network is trained on all data from all classes available at present. First, it is trained on data from just five classes, then from all ten classes and so on, until finally, it is trained from data from all 100 classes simultaneously.

After each increment, the network was trained for 200 epochs, for a total of 4,000 epochs for all 20 increments. We used a learning-rate schedule that resets at the start of each increment. For the first 60 epochs of each increment, the learning rate was set to 0.1, then to 0.02 for the next 60 epochs, then 0.004 for the next 40 epochs and to 0.0008 for the last 40 epochs; we used the initial learning rate and learning-rate schedule reported in ref. 53. During the 200 epochs of training for each increment, we kept track of the network with the best accuracy on the validation set. To prevent overfitting, at the start of each new increment, we reset the weights of the network to the weights of the best-performing (on the validation set) network found during the previous increment; this is equivalent to early stopping for each different increment.

We used an 18-layer deep residual network[38] for all experiments on class-incremental CIFAR-100. The network architecture is described in detail in Extended Data Table 1. The weights of convolutional and linear layers were initialized using Kaiming initialization[54], the weights for the batch-norm layers were initialized to one and all of the bias terms in the network were initialized to zero. Each time five new classes were made available to the network, five more output units were added to the final layer of the network. The weights and biases of these output units were initialized using the same initialization scheme. The weights of the network were optimized using SGD with a momentum of 0.9, a weight decay of 0.0005 and a mini-batch size of 90.

We used several steps of data preprocessing before the images were presented to the network. First, the value of all the pixels in each image was rescaled between 0 and 1 through division by 255. Then, each pixel in each channel was centred and rescaled by the average and standard deviation of the pixel values of each channel, respectively. Finally, we applied three random data transformations to each image before feeding it to the network: randomly horizontally flip the image with a probability of 0.5, randomly crop the image by padding the image with 4 pixels on each side and randomly cropping to the original size, and randomly rotate the image between 0 and 15°. The first two steps of preprocessing were applied to the training, validation and test sets, but the random transformations were only applied to the images in the training set.

We tested several hyperparameters to ensure the best performance for each different algorithm with our specific architecture. For the base system, we tested values for the weight decay parameter in {0.005, 0.0005, 0.00005}. A weight-decay value of 0.0005 resulted in the best performance in terms of area under the curve for accuracy on the test set over the 20 increments. For Shrink and Perturb, we used the weight-decay value of the base system and tested values for the standard deviation of the Gaussian noise in {$10^{-4}$, $10^{-5}$, $10^{-6}$}; $10^{-5}$ resulted in the best performance. For continual backpropagation, we tested values for the maturity threshold in {1,000, 10,000} and for the replacement rate in {$10^{-4}$, $10^{-5}$, $10^{-6}$} using the contribution utility described in equation (1). A maturity threshold of 1,000 and a replacement rate of $10^{-5}$ resulted in the best performance. Finally, for the head-resetting baseline, in Extended Data Fig. 1a, we used the same hyperparameters as for the base system, but the output layer was reinitialized at the start of each increment.

In Fig. 2d, we plot the stable rank of the representation in the penultimate layer of the network and the percentage of dead units in the full network. For a matrix $\mathbf{\Phi} \in \mathbb{R}^{n \times m}$ with singular values $\sigma_k$ sorted in descending order for $k = 1, 2, ..., q$ and $q = \max(n, m)$, the stable rank[55] is $\min\left\{k : \frac{\Sigma_i^k \sigma_i}{\Sigma_j^q \sigma_j} > 0.99\right\}$.

For reference, we also implemented a network with the same hyperparameters as the base system but that was reinitialized at the beginning of each increment. Figure 2b shows the performance of each algorithm relative to the performance of the reinitialized network. For completeness, Extended Data Fig. 1a shows the test accuracy of each algorithm in each different increment. The final accuracy of continual backpropagation on all 100 classes was 76.13%, whereas Extended Data Fig. 1b shows the performance of continual backpropagation for different replacement rates with a maturity threshold of 1,000. For all algorithms that we tested, there was no correlation between when a class was presented and the accuracy of that class, implying that the temporal order of classes did not affect performance.

## Robust loss of plasticity in permuted MNIST

We now use a computationally cheap problem based on the MNIST dataset[56] to test the generality of loss of plasticity across various conditions. MNIST is one of the most common supervised-learning datasets used in deep learning. It consists of 60,000, 28 × 28, greyscale images of handwritten digits from 0 to 9, together with their correct labels. For example, the left image in Extended Data Fig. 3a shows an image that is labelled by the digit 7. The smaller number of classes and the simpler images enable much smaller networks to perform well on this dataset than are needed on ImageNet or CIFAR-100. The smaller networks in turn mean that much less computation is needed to perform the experiments and thus experiments can be performed in greater quantities and under a variety of different conditions, enabling us to perform deeper and more extensive studies of plasticity.

We created a continual supervised-learning problem using permuted MNIST datasets[57,58]. An individual permuted MNIST dataset is created by permuting the pixels in the original MNIST dataset. The right image in Extended Data Fig. 3a is an example of such a permuted image. Given a way of permuting, all 60,000 images are permuted in the same way to produce the new permuted MNIST dataset. Furthermore, we normalized pixel values between 0 and 1 by dividing by 255.

By repeatedly randomly selecting from the approximately $10^{1930}$ possible permutations, we created a sequence of 800 permuted MNIST datasets and supervised-learning tasks. For each task, we presented each of its 60,000 images one by one in random order to the learning network. Then we moved to the next permuted MNIST task and repeated the whole procedure, and so on for up to 800 tasks. No indication was given to the network at the time of task switching. With the pixels being permuted in a completely unrelated way, we might expect classification performance to fall substantially at the time of each task switch. Nevertheless, across tasks, there could be some savings, some improvement in speed of learning or, alternatively, there could be loss of plasticity—loss of the ability to learn across tasks. The network was trained on a single pass through the data and there were no mini-batches. We call this problem Online Permuted MNIST.

We applied feed-forward neural networks with three hidden layers to Online Permuted MNIST. We did not use convolutional layers, as they could not be helpful on the permuted problem because the

spatial information is lost; in MNIST, convolutional layers are often not used even on the standard, non-permuted problem. For each example, the network estimated the probabilities of each of the item classes, compared them to the correct label and performed SGD on the cross-entropy loss. As a measure of online performance, we recorded the percentage of times the network correctly classified each of the 60,000 images in the task. We plot this per-task performance measure versus task number in Extended Data Fig. 3b. The weights were initialized according to a Kaiming distribution.

The left panel of Extended Data Fig. 3b shows the progression of online performance across tasks for a network with 2,000 units per layer and various values of the step-size parameter. Note that that performance first increased across tasks, then began falling steadily across all subsequent tasks. This drop in performance means that the network is slowly losing the ability to learn from new tasks. This loss of plasticity is consistent with the loss of plasticity that we observed in ImageNet and CIFAR-100.

Next, we varied the network size. Instead of 2,000 units per layer, we tried 100, 1,000 and 10,000 units per layer. We ran this experiment for only 150 tasks, primarily because the largest network took much longer to run. The performances at good step sizes for each network size are shown in the middle panel of Extended Data Fig. 3b. Loss of plasticity with continued training is most pronounced at the smaller network sizes, but even the largest networks show some loss of plasticity.

Next, we studied the effect of the rate at which the task changed. Going back to the original network with 2,000-unit layers, instead of changing the permutation after each 60,000 examples, we now changed it after each 10,000, 100,000 or 1 million examples and ran for 48 million examples in total no matter how often the task changed. The examples in these cases were selected randomly with replacement for each task. As a performance measure of the network on a task, we used the percentage correct over all of the images in the task. The progression of performance is shown in the right panel in Extended Data Fig. 3b. Again, performance fell across tasks, even if the change was very infrequent. Altogether, these results show that the phenomenon of loss of plasticity robustly arises in this form of backpropagation. Loss of plasticity happens for a wide range of step sizes, rates of distribution change and for both underparameterized and overparameterized networks.

## Loss of plasticity with different activations in the Slowly-Changing Regression problem

There remains the issue of the network's activation function. In our experiments so far, we have used ReLU, the most popular choice at present, but there are several other possibilities. For these experiments, we switched to an even smaller, more idealized problem. Slowly-Changing Regression is a computationally inexpensive problem in which we can run a single experiment on a CPU core in 15 min, allowing us to perform extensive studies. As its name suggests, this problem is a regression problem—meaning that the labels are real numbers, with a squared loss, rather than nominal values with a cross-entropy loss—and the non-stationarity is slow and continual rather than abrupt, as in a switch from one task to another. In Slowly-Changing Regression, we study loss of plasticity for networks with six popular activation functions: sigmoid, tanh, ELU[59], leaky ReLU[60], ReLU[61] and Swish[62].

In Slowly-Changing Regression, the learner receives a sequence of examples. The input for each example is a binary vector of size $m + 1$. The input has $f$ slow-changing bits, $m - f$ random bits and then one constant bit. The first $f$ bits in the input vector change slowly. After every $T$ examples, one of the first $f$ bits is chosen uniformly at random and its value is flipped. These first $f$ bits remain fixed for the next $T$ examples. The parameter $T$ allows us to control the rate at which the input distribution changes. The next $m - f$ bits are randomly sampled for each example. Last, the $(m + 1)$th bit is a bias term with a constant value of one.

The target output is generated by running the input vector through a neural network, which is set at the start of the experiment and kept fixed. As this network generates the target output and represents the desired solution, we call it the target network. The weights of the target networks are randomly chosen to be +1 or −1. The target network has one hidden layer with the linear threshold unit (LTU) activation. The value of the $i$th LTU is one if the input is above a threshold $\theta_i$ and 0 otherwise. The threshold $\theta_i$ is set to be equal to $(m + 1) \times \beta - S_i$, in which $\beta \in [0, 1]$ and $S_i$ is the number of input weights with negative value[63]. The details of the input and target function in the Slowly-Changing Regression problem are also described in Extended Data Fig. 2a.

The details of the specific instance of the Slowly-Changing Regression problem we use in this paper and the learning network used to predict its output are listed in Extended Data Table 4. Note that the target network is more complex than the learning network, as the target network is wider, with 100 hidden units, whereas the learner has just five hidden units. Thus, because the input distribution changes every $T$ example and the target function is more complex than what the learner can represent, there is a need to track the best approximation.

We applied learning networks with different activation functions to the Slowly-Changing Regression. The learner used the backpropagation algorithm to learn the weights of the network. We used a uniform Kaiming distribution[54] to initialize the weights of the learning network. The distribution is $U(-b, b)$ with bound, $b = \text{gain} \times \sqrt{\frac{3}{\text{num\_inputs}}}$, in which gain is chosen such that the magnitude of inputs does not change across layers. For tanh, sigmoid, ReLU and leaky ReLU, the gain is 5/3, 1, $\sqrt{2}$ and $\sqrt{2/(1 + \alpha^2)}$, respectively. For ELU and Swish, we used gain = $\sqrt{2}$, as was done in the original papers[59,62].

We ran the experiment on the Slowly-Changing Regression problem for 3 million examples. For each activation and value of step size, we performed 100 independent runs. First, we generated 100 sequences of examples (input–output pairs) for the 100 runs. Then these 100 sequences of examples were used for experiments with all activations and values of the step-size parameter. We used the same sequence of examples to control the randomness in the data stream across activations and step sizes.

The results of the experiments are shown in Extended Data Fig. 2b. We measured the squared error, that is, the square of the difference between the true target and the prediction made by the learning network. In Extended Data Fig. 2b, the squared error is presented in bins of 40,000 examples. This means that the first data point is the average squared error on the first 40,000 examples, the next is the average squared error on the next 40,000 examples and so on. The shaded region in the figure shows the standard error of the binned error.

Extended Data Fig. 2b shows that, in Slowly-Changing Regression, after performing well initially, the error increases for all step sizes and activations. For some activations such as ReLU and tanh, loss of plasticity is severe, and the error increases to the level of the linear baseline. Although for other activations such as ELU loss of plasticity is less severe, there is still a notable loss of plasticity. These results mean that loss of plasticity is not resolved by using commonly used activations. The results in this section complement the results in the rest of the article and add to the generality of loss of plasticity in deep learning.

## Understanding loss of plasticity

We now turn our attention to understanding why backpropagation loses plasticity in continual-learning problems. The only difference in the learner over time is the network weights. In the beginning, the weights were small random numbers, as they were sampled from the initial distribution; however, after learning some tasks, the weights became optimized for the most recent task. Thus, the starting weights for the next task are qualitatively different from those for the first task. As this difference in the weights is the only difference in the learning algorithm over time, the initial weight distribution must have some

unique properties that make backpropagation plastic in the beginning. The initial random distribution might have many properties that enable plasticity, such as the diversity of units, non-saturated units, small weight magnitude etc.

As we now demonstrate, many advantages of the initial distribution are lost concurrently with loss of plasticity. The loss of each of these advantages partially explains the degradation in performance that we have observed. We then provide arguments for how the loss of these advantages could contribute to loss of plasticity and measures that quantify the prevalence of each phenomenon. We provide an in-depth study of the Online Permuted MNIST problem that will serve as motivation for several solution methods that could mitigate loss of plasticity.

The first noticeable phenomenon that occurs concurrently with the loss of plasticity is the continual increase in the fraction of constant units. When a unit becomes constant, the gradients flowing back from the unit become zero or very close to zero. Zero gradients mean that the weights coming into the unit do not change, which means that this unit loses all of its plasticity. In the case of ReLU activations, this occurs when the output of the activations is zero for all examples of the task; such units are often said to be dead[64,65]. In the case of the sigmoidal activation functions, this phenomenon occurs when the output of a unit is too close to either of the extreme values of the activation function; such units are often said to be saturated[66,67].

To measure the number of dead units in a network with ReLU activation, we count the number of units with a value of zero for all examples in a random sample of 2,000 images at the beginning of each new task. An analogous measure in the case of sigmoidal activations is the number of units that are $\epsilon$ away from either of the extreme values of the function for some small positive $\epsilon$ (ref. 68). We only focus on ReLU networks in this section.

In our experiments on the Online Permuted MNIST problem, the deterioration of online performance is accompanied by a large increase in the number of dead units (left panel of Extended Data Fig. 3c). For the step size of 0.01, up to 25% of units die after 800 tasks. In the permuted MNIST problem, in which all inputs are positive because they are normalized between 0 and 1, once a unit in the first layer dies, it stays dead forever. Thus, an increase in dead units directly decreases the total capacity of the network. In the next section, we will see that methods that stop the units from dying can substantially reduce loss of plasticity. This further supports the idea that the increase in dead units is one of the causes of loss of plasticity in backpropagation.

Another phenomenon that occurs with loss of plasticity is the steady growth of the network's average weight magnitude. We measure the average magnitude of the weights by adding up their absolute values and dividing by the total number of weights in the network. In the permuted MNIST experiment, the degradation of online classification accuracy of backpropagation observed in Extended Data Fig. 3b is associated with an increase in the average magnitude of the weights (centre panel of Extended Data Fig. 3c). The growth of the magnitude of the weights of the network can represent a problem because large weight magnitudes are often associated with slower learning. The weights of a neural network are directly linked to the condition number of the Hessian matrix in the second-order Taylor approximation of the loss function. The condition number of the Hessian is known to affect the speed of convergence of SGD algorithms (see ref. 69 for an illustration of this phenomenon in convex optimization). Consequently, the growth in the magnitude of the weights could lead to an ill-conditioned Hessian matrix, resulting in a slower convergence.

The last phenomenon that occurs with the loss of plasticity is the drop in the effective rank of the representation. Similar to the rank of a matrix, which represents the number of linearly independent dimensions, the effective rank takes into consideration how each dimension influences the transformation induced by a matrix[70]. A high effective rank indicates that most of the dimensions of the matrix contribute similarly to the transformation induced by the matrix. On the other hand, a low effective rank corresponds to most dimensions having no notable effect on the transformation, implying that the information in most of the dimensions is close to being redundant.

Formally, consider a matrix $\Phi \in \mathbb{R}^{n \times m}$ with singular values $\sigma_k$ for $k = 1, 2, \ldots, q$, and $q = \max(n, m)$. Let $p_k = \sigma_k / \|\sigma\|_1$, in which $\sigma$ is the vector containing all the singular values and $\|\cdot\|_1$ is the $\ell^1$ norm. The effective rank of matrix $\Phi$, or erank($\Phi$), is defined as

$$erank(\Phi) \doteq \exp\{H(p_1, p_2, \ldots, p_q)\},$$

$$\text{in which } H(p_1, p_2, \ldots, p_q) = -\sum_{k=1}^{q} p_k \log(p_k). \qquad (2)$$

Note that the effective rank is a continuous measure that ranges between one and the rank of matrix $\Phi$.

In the case of neural networks, the effective rank of a hidden layer measures the number of units that can produce the output of the layer. If a hidden layer has a low effective rank, then a small number of units can produce the output of the layer, meaning that many of the units in the hidden layer are not providing any useful information. We approximate the effective rank on a random sample of 2,000 examples before training on each task.

In our experiments, loss of plasticity is accompanied by a decrease in the average effective rank of the network (right panel of Extended Data Fig. 3c). This phenomenon in itself is not necessarily a problem. After all, it has been shown that gradient-based optimization seems to favour low-rank solutions through implicit regularization of the loss function or implicit minimization of the rank itself[71,72]. However, a low-rank solution might be a bad starting point for learning from new observations because most of the hidden units provide little to no information.

The decrease in effective rank could explain the loss of plasticity in our experiments in the following way. After each task, the learning algorithm finds a low-rank solution for the current task, which then serves as the initialization for the next task. As the process continues, the effective rank of the representation layer keeps decreasing after each task, limiting the number of solutions that the network can represent immediately at the start of each new task.

In this section, we looked deeper at the networks that lost plasticity in the Online Permuted MNIST problem. We noted that the only difference in the learning algorithm over time is the weights of the network, which means that the initial weight distribution has some properties that allowed the learning algorithm to be plastic in the beginning. And as learning progressed, the weights of the network moved away from the initial distribution and the algorithm started to lose plasticity. We found that loss of plasticity is correlated with an increase in weight magnitude, a decrease in the effective rank of the representation and an increase in the fraction of dead units. Each of these correlates partially explains loss of plasticity faced by backpropagation.

## Existing deep-learning methods for mitigating loss of plasticity

We now investigate several existing methods and test how they affect loss of plasticity. We study five existing methods: L2 regularization[73], Dropout[74], online normalization[75], Shrink and Perturb[11] and Adam[43]. We chose L2 regularization, Dropout, normalization and Adam because these methods are commonly used in deep-learning practice. Although Shrink and Perturb is not a commonly used method, we chose it because it reduces the failure of pretraining, a problem that is an instance of loss of plasticity. To assess if these methods can mitigate loss of plasticity, we tested them on the Online Permuted MNIST problem using the same network architecture we used in the previous section, 'Understanding loss of plasticity'. Similar to the previous section, we measure the online classification accuracy on all 60,000 examples of the task. All the algorithms used a step size of 0.003, which was the best-performing step size for backpropagation in the left panel of Extended Data Fig. 3b. We also use the three correlates of loss of plasticity found in the previous

section to get a deeper understanding of the performance of these methods.

An intuitive way to address loss of plasticity is to use weight regularization, as loss of plasticity is associated with a growth of weight magnitudes, shown in the previous section. We used L2 regularization, which adds a penalty to the loss function proportional to the $\ell^2$ norm of the weights of the network. The L2 regularization penalty incentivizes SGD to find solutions that have a low weight magnitude. This introduces a hyperparameter $\lambda$ that modulates the contribution of the penalty term.

The purple line in the left panel of Extended Data Fig. 4a shows the performance of L2 regularization on the Online Permuted MNIST problem. The purple lines in the other panels of Extended Data Fig. 4a show the evolution of the three correlates of loss of plasticity with L2 regularization. For L2 regularization, the weight magnitude does not continually increase. Moreover, as expected, the non-increasing weight magnitude is associated with lower loss of plasticity. However, L2 regularization does not fully mitigate loss of plasticity. The other two correlates for loss of plasticity explain this, as the percentage of dead units kept increasing and the effective rank kept decreasing. Finally, Extended Data Fig. 4b shows the performance of L2 regularization for different values of $\lambda$. The regularization parameter $\lambda$ controlled the peak of the performance and how quickly it decreased.

A method related to weight regularization is Shrink and Perturb[11]. As the name suggests, Shrink and Perturb performs two operations; it shrinks all the weights and then adds random Gaussian noise to these weights. The introduction of noise introduces another hyperparameter, the standard deviation of the noise. Owing to the shrinking part of Shrink and Perturb, the algorithm favours solutions with smaller average weight magnitude than backpropagation. Moreover, the added noise prevents units from dying because it adds a non-zero probability that a dead unit will become active again. If Shrink and Perturb mitigates these correlates to loss of plasticity, it could reduce loss of plasticity.

The performance of Shrink and Perturb is shown in orange in Extended Data Fig. 4. Similar to L2 regularization, Shrink and Perturb stops the weight magnitude from continually increasing. Moreover, it also reduces the percentage of dead units. However, it has a lower effective rank than backpropagation, but still higher than that of L2 regularization. Not only does Shrink and Perturb have a lower loss of plasticity than backpropagation but it almost completely mitigates loss of plasticity in Online Permuted MNIST. However, Shrink and Perturb was sensitive to the standard deviation of the noise. If the noise was too high, loss of plasticity was much more severe, and if it was too low, it did not have any effect.

An important technique in modern deep learning is called Dropout[74]. Dropout randomly sets each hidden unit to zero with a small probability, which is a hyperparameter of the algorithm. The performance of Dropout is shown in pink in Extended Data Fig. 4.

Dropout showed similar measures of percentage of dead units, weight magnitude and effective rank as backpropagation, but, surprisingly, showed higher loss of plasticity. The poor performance of Dropout is not explained by our three correlates of loss of plasticity, which means that there are other possible causes of loss of plasticity. A thorough investigation of Dropout is beyond the scope of this paper, though it would be an interesting direction for future work. We found that a higher Dropout probability corresponded to a faster and sharper drop in performance. Dropout with probability of 0.03 performed the best and its performance was almost identical to that of backpropagation. However, Extended Data Fig. 4a shows the performance for a Dropout probability of 0.1 because it is more representative of the values used in practice.

Another commonly used technique in deep learning is batch normalization[76]. In batch normalization, the output of each hidden layer is normalized and rescaled using statistics computed from each mini-batch

of data. We decided to include batch normalization in this investigation because it is a popular technique often used in practice. Because batch normalization is not amenable to the online setting used in the Online Permuted MNIST problem, we used online normalization[77] instead, an online variant of batch normalization. Online normalization introduces two hyperparameters used for the incremental estimation of the statistics in the normalization steps.

The performance of online normalization is shown in green in Extended Data Fig. 4. Online normalization had fewer dead units and a higher effective rank than backpropagation in the earlier tasks, but both measures deteriorated over time. In the later tasks, the network trained using online normalization has a higher percentage of dead units and a lower effective rank than the network trained using backpropagation. The online classification accuracy is consistent with these results. Initially, it has better classification accuracy, but later, its classification accuracy becomes lower than that of backpropagation. For online normalization, the hyperparameters changed when the performance of the method peaked, and it also slightly changed how fast it got to its peak performance.

No assessment of alternative methods can be complete without Adam[43], as it is considered one of the most useful tools in modern deep learning. The Adam optimizer is a variant of SGD that uses an estimate of the first moment of the gradient scaled inversely by an estimate of the second moment of the gradient to update the weights instead of directly using the gradient. Because of its widespread use and success in both supervised and reinforcement learning, we decided to include Adam in this investigation to see how it would affect the plasticity of deep neural networks. Adam has two hyperparameters that are used for computing the moving averages of the first and second moments of the gradient. We used the default values of these hyperparameters proposed in the original paper and tuned the step-size parameter.

The performance of Adam is shown in cyan in Extended Data Fig. 4. Adam's loss of plasticity can be categorized as catastrophic, as it reduces substantially. Consistent with our previous results, Adam scores poorly in the three measures corresponding to the correlates of loss of plasticity. Adam had an early increase in the percentage of dead units that plateaus at around 60%, similar weight magnitude as backpropagation and a large drop in the effective rank early during training. We also tested Adam with different activation functions on the Slowly-Changing Regression and found that loss of plasticity with Adam is usually worse than with SGD.

Many of the standard methods substantially worsened loss of plasticity. The effect of Adam on the plasticity of the networks was particularly notable. Networks trained with Adam quickly lost almost all of their diversity, as measured by the effective rank, and gained a large percentage of dead units. This marked loss of plasticity of Adam is an important result for deep reinforcement learning, for which Adam is the default optimizer[78], and reinforcement learning is inherently continual owing to the ever-changing policy. Similar to Adam, other commonly used methods such as Dropout and normalization worsened loss of plasticity. Normalization had better performance in the beginning, but later it had a sharper drop in performance than backpropagation. In the experiment, Dropout simply made the performance worse. We saw that the higher the Dropout probability, the larger the loss of plasticity. These results mean that some of the most successful tools in deep learning do not work well in continual learning, and we need to focus on directly developing tools for continual learning.

We did find some success in maintaining plasticity in deep neural networks. L2 regularization and Shrink and Perturb reduce loss of plasticity. Shrink and Perturb is particularly effective, as it almost entirely mitigates loss of plasticity. However, both Shrink and Perturb and L2 regularization are slightly sensitive to hyperparameter values. Both methods only reduce loss of plasticity for a small range of hyperparameters, whereas for other hyperparameter values, they make loss of plasticity worse. This sensitivity to hyperparameters can limit the

application of these methods to continual learning. Furthermore, Shrink and Perturb does not fully resolve the three correlates of loss of plasticity, it has a lower effective rank than backpropagation and it still has a high fraction of dead units.

We also applied continual backpropagation on Online Permuted MNIST. The replacement rate is the main hyperparameter in continual backpropagation, as it controls how rapidly units are reinitialized in the network. For example, a replacement rate of $10^{-6}$ for our network with 2,000 hidden units in each layer would mean replacing one unit in each layer after every 500 examples.

Blue lines in Extended Data Fig. 4 show the performance of continual backpropagation. It has a non-degrading performance and is stable for a wide range of replacement rates. Continual backpropagation also mitigates all three correlates of loss of plasticity. It has almost no dead units, stops the network weights from growing and maintains a high effective rank across tasks. All algorithms that maintain a low weight magnitude also reduced loss of plasticity. This supports our claim that low weight magnitudes are important for maintaining plasticity. The algorithms that maintain low weight magnitudes were continual backpropagation, L2 regularization and Shrink and Perturb. Shrink and Perturb and continual backpropagation have an extra advantage over L2 regularization: they inject randomness into the network. This injection of randomness leads to a higher effective rank and lower number of dead units, which leads to both of these algorithms performing better than L2 regularization. However, continual backpropagation injects randomness selectively, effectively removing all dead units from the network and leading to a higher effective rank. This smaller number of dead units and a higher effective rank explains the better performance of continual backpropagation.

## Details and further analysis in reinforcement learning

The experiments presented in the main text were conducted using the Ant-v3 environment from OpenAI Gym[79]. We changed the coefficient of friction by sampling it log-uniformly from the range [0.02, 2.00], using a logarithm with base 10. The coefficient of friction changed at the first episode boundary after 2 million time steps had passed since the last change. We also tested Shrink and Perturb on this problem and found that it did not provide a marked performance improvement over L2 regularization. Two separate networks were used for the policy and the value function, and both had two hidden layers with 256 units. These networks were trained using Adam alongside PPO to update the weights in the network. See Extended Data Table 5 for the values of the other hyperparameters. In all of the plots showing results of reinforcement-learning experiments, the shaded region represents the 95% bootstrapped confidence[80].

The reward signal in the ant problem consists of four components. The main component rewards the agent for forward movement. It is proportional to the distance moved by the ant in the positive $x$ direction since the last time step. The second component has a value of 1 at each time step. The third component penalizes the ant for taking large actions. This component is proportional to the square of the magnitude of the action. Finally, the last component penalizes the agent for large external contact forces. It is proportional to the sum of external forces (clipped in a range). The reward signal at each time step is the sum of these four components.

We also evaluated PPO and its variants in two more environments: Hopper-v3 and Walker-v3. The results for these experiments are presented in Extended Data Fig. 5a. The results mirrored those from Ant-v3; standard PPO suffered from a notable degradation in performance, in which its performance decreased substantially. However, this time, L2 regularization did not fix the issue in all cases; there was some performance degradation with L2 in Walker-v3. PPO, with continual backpropagation and L2 regularization, completely fixed the issue in all environments. Note that the only difference between our experiments and what is typically done in the literature is that we run the experiments

for longer. Typically, these experiments are only done for 3 million steps, but we ran these experiments for up to 100 million steps.

PPO with L2 regularization only avoided degradation for a relatively large value of weight decay, $10^{-3}$. This extreme regularization stops the agent from finding better policies and stays stuck at a suboptimal policy. There was large performance degradation for smaller values of weight decay, and for larger values, the performance was always low. When we used continual backpropagation and L2 regularization together, we could use smaller values of weight decay. All the results for PPO with continual backpropagation and L2 regularization have a weight decay of $10^{-4}$, a replacement rate of $10^{-4}$ and a maturity threshold of $10^4$. We found that the performance of PPO with continual backpropagation and L2 regularization was sensitive to the replacement rate but not to the maturity threshold and weight decay.

PPO uses the Adam optimizer, which keeps running estimates of the gradient and the squared of the gradient. These estimates require two further parameters, called $\beta_1$ and $\beta_2$. The standard values of $\beta_1$ and $\beta_2$ are 0.9 and 0.999, respectively, which we refer to as standard Adam. Lyle et al.[24] showed that the standard values of $\beta_1$ and $\beta_2$ cause a large loss of plasticity. This happens because of the mismatch in $\beta_1$ and $\beta_2$. A sudden large gradient can cause a very large update, as a large value of $\beta_2$ means that the running estimate for the square of the gradient, which is used in the denominator, is updated much more slowly than the running estimate for the gradient, which is the numerator. This loss of plasticity in Adam can be reduced by setting $\beta_1$ equal to $\beta_2$. In our experiments, we set $\beta_1$ and $\beta_2$ to 0.99 and refer to it as tuned Adam/PPO. In Extended Data Fig. 5c, we measure the largest total weight change in the network during a single update cycle for bins of 1 million steps. The first point in the plots shows the largest weight change in the first 1 million steps. The second point shows the largest weight change in the second 1 second steps and so on. The figure shows that standard Adam consistently causes very large updates to the weights, which can destabilize learning, whereas tuned Adam with $\beta_1 = \beta_2 = 0.99$ has substantially smaller updates, which leads to more stable learning. In all of our experiments, all algorithms other than the standard PPO used the tuned parameters for Adam ($\beta_1 = \beta_2 = 0.99$). The failure of standard Adam with PPO is similar to the failure of standard Adam in permuted MNIST.

In our next experiment, we perform a preliminary comparison with ReDo[25]. ReDo is another selective reinitialization method that builds on continual backpropagation but uses a different measure of utility and strategy for reinitializing. We tested ReDo on Ant-v3, the hardest of the three environments. ReDo requires two parameters: a threshold and a reinitialization period. We tested ReDo for all combinations of thresholds in {0.01, 0.03, 0.1} and reinitialization periods in {10, $10^2$, $10^3$, $10^4$, $10^5$}; a threshold of 0.1 with a reinitialization period of $10^2$ performed the best. The performance of PPO with ReDo is plotted in Extended Data Fig. 5b. ReDo and continual backpropagation were used with weight decay of $10^{-4}$ and $\beta_1$ and $\beta_2$ of 0.99. The figure shows that PPO with ReDo and L2 regularization performs much better than standard PPO. However, it still suffers from performance degradation and its performance is worse than PPO with L2 regularization. Note that this is only a preliminary comparison; we leave a full comparison and analysis of both methods for future work.

The performance drop of PPO in stationary environments is a nuanced phenomenon. Loss of plasticity and forgetting are both responsible for the observed degradation in performance. The degradation in performance implies that the agent forgot the good policy it had once learned, whereas the inability of the agent to relearn a good policy means it lost plasticity.

Loss of plasticity expresses itself in various forms in deep reinforcement learning. Some work found that deep reinforcement learning systems can lose their generalization abilities in the presence of non-stationarities[81]. A reduction in the effective rank, similar to the rank reduction in CIFAR-100, has been observed in some deep

reinforcement-learning algorithms[82]. Nikishin et al.[18] showed that many reinforcement-learning systems perform better if their network is occasionally reset to its naive initial state, retaining only the replay buffer. This is because the learning networks became worse than a reinitialized network at learning from new data. Recent work has improved performance in many reinforcement-learning problems by applying plasticity-preserving methods[25,83–87]. These works focused on deep reinforcement learning systems that use large replay buffers. Our work complements this line of research as we studied systems based on PPO, which has much smaller replay buffers. Loss of plasticity is most relevant for systems that use small or no replay buffers, as large buffers can hide the effect of new data. Overcoming loss of plasticity is an important step towards deep reinforcement-learning systems that can learn from an online data stream.

## Extended discussion

There are two main goals in continual learning: maintaining stability and maintaining plasticity[88–91]. Maintaining stability is concerned with memorizing useful information and maintaining plasticity is about finding new useful information when the data distribution changes. Current deep-learning methods struggle to maintain stability as they tend to forget previously learned information[28,29]. Many papers have been dedicated to maintaining stability in deep continual learning[30,92–97]. We focused on continually finding useful information, not on remembering useful information. Our work on loss of plasticity is different but complementary to the work on maintaining stability. Continual backpropagation in its current form does not tackle the forgetting problem. Its current utility measure only considers the importance of units for current data. One idea to tackle forgetting is to use a long-term measure of utility that remembers which units were useful in the past. Developing methods that maintain both stability and plasticity is an important direction for future work.

There are many desirable properties for an efficient continual-learning system[98,99]. It should be able to keep learning new things, control what it remembers and forgets, have good computational and memory efficiency and use previous knowledge to speed up learning on new data. The choice of the benchmark affects which property is being focused on. Most benchmarks and evaluations in our paper only focused on plasticity but not on other aspects, such as forgetting and speed of learning. For example, in Continual ImageNet, previous tasks are rarely repeated, which makes it effective for studying plasticity but not forgetting. In permuted MNIST, consecutive tasks are largely independent, which makes it suitable for studying plasticity in isolation. However, this independence means that previous knowledge cannot substantially speed up learning on new tasks. On the other hand, in class-incremental CIFAR-100, previous knowledge can substantially speed up learning of new classes. Overcoming loss of plasticity is an important, but still the first, step towards the goal of fast learning on future data[100–102]. Once we have networks that maintain plasticity, we can develop methods that use previous knowledge to speed up learning on future data.

Loss of plasticity is a critical factor when learning continues for many tasks, but it might be less important if learning happens for a small number of tasks. Usually, the learning system can take advantage of previous learning in the first few tasks. For example, in class-incremental CIFAR-100 (Fig. 2), the base deep-learning systems performed better than the network trained from scratch for up to 40 classes. This result is consistent with deep-learning applications in which the learning system is first trained on a large dataset and then fine-tuned on a smaller, more relevant dataset. Plasticity-preserving methods such as continual backpropagation may still improve performance in such applications based on fine-turning, but we do not expect that improvement to be large, as learning happens only for a small number of tasks. We have observed that deep-learning systems gradually lose plasticity, and this effect accumulates over tasks. Loss of plasticity becomes an important factor when learning continues for a large number of tasks; in class-incremental CIFAR-100, the performance of the base deep-learning system was much worse after 100 classes.

We have made notable progress in understanding loss of plasticity. However, it remains unclear which specific properties of initialization with small random numbers are important for maintaining plasticity. Recent work[103,104] has made exciting progress in this direction and it remains an important avenue for future work. The type of loss of plasticity studied in this article is largely because of the loss of the ability to optimize new objectives. This is different from the type of loss of plasticity in which the system can keep optimizing new objectives but lose the ability to generalize[11,12]. However, it is unclear if the two types of plasticity loss are fundamentally different or if the same mechanism can explain both phenomena. Future work that improves our understanding of plasticity and finds the underlying causes of both types of plasticity loss will be valuable to the community.

Continual backpropagation uses a utility measure to find and replace low-utility units. One limitation of continual backpropagation is that the utility measure is based on heuristics. Although it performs well, future work on more principled utility measures will improve the foundations of continual backpropagation. Our current utility measure is not a global measure of utility as it does not consider how a given unit affects the overall represented function. One possibility is to develop utility measures in which utility is propagated backwards from the loss function. The idea of utility in continual backpropagation is closely related to connection utility in the neural-network-pruning literature. Various papers[105–108] have proposed different measures of connection utility for the network-pruning problem. Adapting these utility measures to mitigate loss of plasticity is a promising direction for new algorithms and some recent work is already making progress in this direction[109].

The idea of selective reinitialization is similar to the emerging idea of dynamic sparse training[110–112]. In dynamic sparse training, a sparse network is trained from scratch and connections between different units are generated and removed during training. Removing connections requires a measure of utility, and the initialization of new connections requires a generator similar to selective reinitialization. The main difference between dynamic sparse training and continual backpropagation is that dynamic sparse training operates on connections between units, whereas continual backpropagation operates on units. Consequently, the generator in dynamic sparse training must also decide which new connections to grow. Dynamic sparse training has achieved promising results in supervised and reinforcement-learning problems[113–115], in which dynamic sparse networks achieve performance close to dense networks even at high sparsity levels. Dynamic sparse training is a promising idea that can be useful to maintain plasticity.

The idea of adding new units to neural networks is present in the continual-learning literature[92,116,117]. This idea is usually manifested in algorithms that dynamically increase the size of the network. For example, one method[117] expands the network by allocating a new subnetwork whenever there is a new task. These methods do not have an upper limit on memory requirements. Although these methods are related to the ideas in continual backpropagation, none are suitable for comparison, as continual backpropagation is designed for learning systems with finite memory, which are well suited for lifelong learning. And these methods would therefore require non-trivial modification to apply to our setting of finite memory.

Previous works on the importance of initialization have focused on finding the correct weight magnitude to initialize the weights. It has been shown that it is essential to initialize the weights so that the gradients do not become exponentially small in the initial layers of a network and the gradient is preserved across layers[54,66]. Furthermore, initialization with small weights is critical for sigmoid activations as they may saturate if the weights are too large[118]. Despite all this work

on the importance of initialization, the fact that its benefits are only present initially but not continually has been overlooked, as these papers focused on cases in which learning has to be done just once, not continually.

Continual backpropagation selectively reinitializes low-utility units. One common strategy to deal with non-stationary data streams is reinitializing the network entirely. In the Online Permuted MNIST experiment, full reinitialization corresponds to a performance that stays at the level of the first point (Extended Data Fig. 4a). In this case, continual backpropagation outperforms full reinitialization as it takes advantage of what it has previously learned to speed up learning on new data. In ImageNet experiments, the final performance of continual backpropagation is only slightly better than a fully reinitialized network (the first point for backpropagation in left panel of Fig. 1b). However, Fig. 1 does not show how fast an algorithm reaches the final performance in each task. We observed that continual backpropagation achieves the best accuracy ten times faster than a fully reinitialized network on the 5,000th task of Continual ImageNet, ten epochs versus about 125 epochs. Furthermore, continual backpropagation could be combined with other methods that mitigate forgetting, which can further speed up learning on new data. In reinforcement learning, full reinitialization is only practical for systems with a large buffer. For systems that keep a small or no buffer, such as those we studied, full reinitialization will lead the agent to forget everything it has learned, and its performance will be down to the starting point.

Loss of plasticity might also be connected to the lottery ticket hypothesis[119]. The hypothesis states that randomly initialized networks contain subnetworks that can achieve performance close to that of the original network with a similar number of updates. These subnetworks are called winning tickets. We found that, in continual-learning problems, the effective rank of the representation at the beginning of tasks reduces over time. In a sense, the network obtained after training on several tasks has less randomness and diversity than the original random network. The reduced randomness might mean that the network has fewer winning tickets. And this reduced number of winning tickets might explain loss of plasticity. Our understanding of loss of plasticity could be deepened by fully exploring its connection with the lottery ticket hypothesis.

Some recent works have focused on quickly adapting to the changes in the data stream[120–122]. However, the problem settings in these papers were offline as they had two separate phases, one for learning and the other for evaluation. To use these methods online, they have to be pretrained on tasks that represent tasks that the learner will encounter during the online evaluation phase. This requirement of having access to representative tasks in the pretraining phase is not realistic for lifelong learning systems as the real world is non-stationary, and even the distribution of tasks can change over time. These methods are not comparable with those we studied in our work, as we studied fully online methods that do not require pretraining.

In this work, we found that methods that continually injected randomness while maintaining small weight magnitudes greatly reduced loss of plasticity. Many works have found that adding noise while training neural networks can improve training and testing performance. The main benefits of adding noise have been reported to be avoiding overfitting and improving training performance[123–125]. However, it can be tricky to inject noise without degrading performance in some cases[126]. In our case, when the data distribution is non-stationary, we found that continually injecting noise along with L2 regularization helps with maintaining plasticity in neural networks.

## Data availability

All of the datasets and simulation environments used in this work are publicly available. Other data needed to evaluate the conclusions in the article are present in the article or the extended data.

## Code availability

The code is available at https://github.com/shibhansh/loss-of-plasticity.

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

**Acknowledgements** We thank M. White for her feedback on an earlier version of this work; P. Nagarajan, E. Graves, G. Mihucz, A. Hakhverdyan, K. Roice, T. Ferguson, L. Watson, H. Sinha, P. Bhangale and M. Przystupa for their feedback on writing; and M. C. Machado for encouraging us to make this work accessible to a general scientific audience. We gratefully acknowledge the Digital Research Alliance of Canada for providing the computational resources to carry out the experiments in this paper. We also acknowledge funding from the Canada CIFAR AI Chairs program, DeepMind, the Alberta Machine Intelligence Institute (Amii), CIFAR and the Natural Sciences and Engineering Research Council of Canada (NSERC). This work was made possible by the stimulating and supportive research environment created by the members of the Reinforcement Learning and Artificial Intelligence (RLAI) laboratory, particularly within the agent-state research meetings.

**Author contributions** S.D., J.F.H.-G., Q.L. and A.R.M. wrote the software. S.D., J.F.H.-G. and P.R. prepared the datasets. S.D. and J.F.H.-G. designed the experiments. S.D., J.F.H.-G., Q.L., R.S.S. and A.R.M. analysed and interpreted the results. S.D., A.R.M. and R.S.S. developed the continual backpropagation algorithm. S.D., J.F.H.-G., Q.L., R.S.S. and A.R.M. prepared the manuscript.

**Competing interests** The authors declare no competing interests.

**Additional information**
**Correspondence and requests for materials** should be addressed to Shibhansh Dohare.

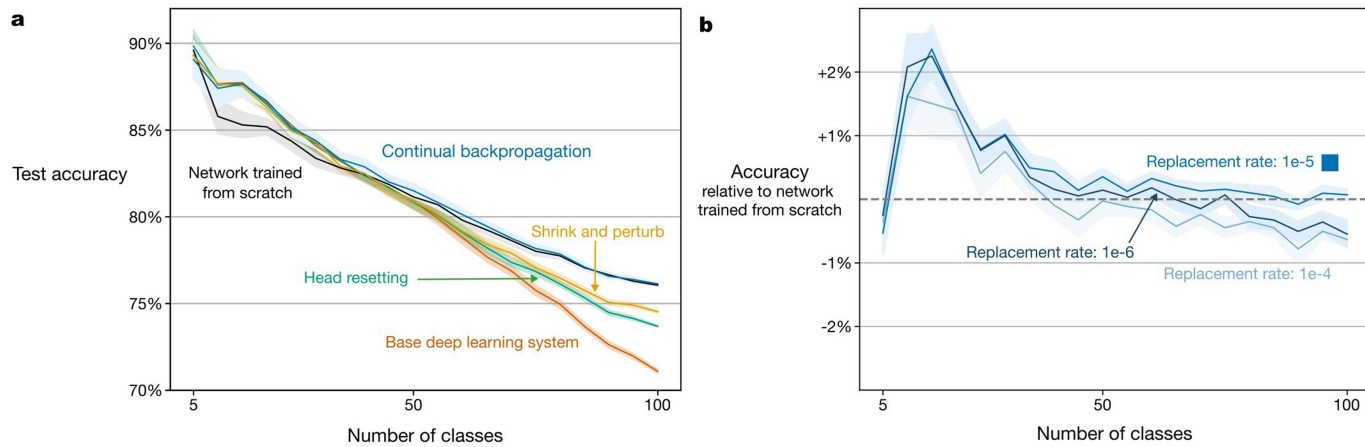

**Extended Data Fig. 1 | Further results on class-incremental CIFAR-100.**
**a**, Test accuracy in class-incremental CIFAR-100. As more classes are added,
the classification becomes harder and algorithms naturally show decreasing
accuracy with more classes. Each line corresponds to the average of 15 runs.
**b**, Test accuracy of continual backpropagation for different values of the
replacement rate parameter with contribution utility and 1,000 maturity
threshold. The line corresponding to $10^{-4}$ is an average of five runs, whereas the
other two lines are an average of 15 runs. The solid lines represent the mean and
the shaded regions correspond to ±1 standard error.

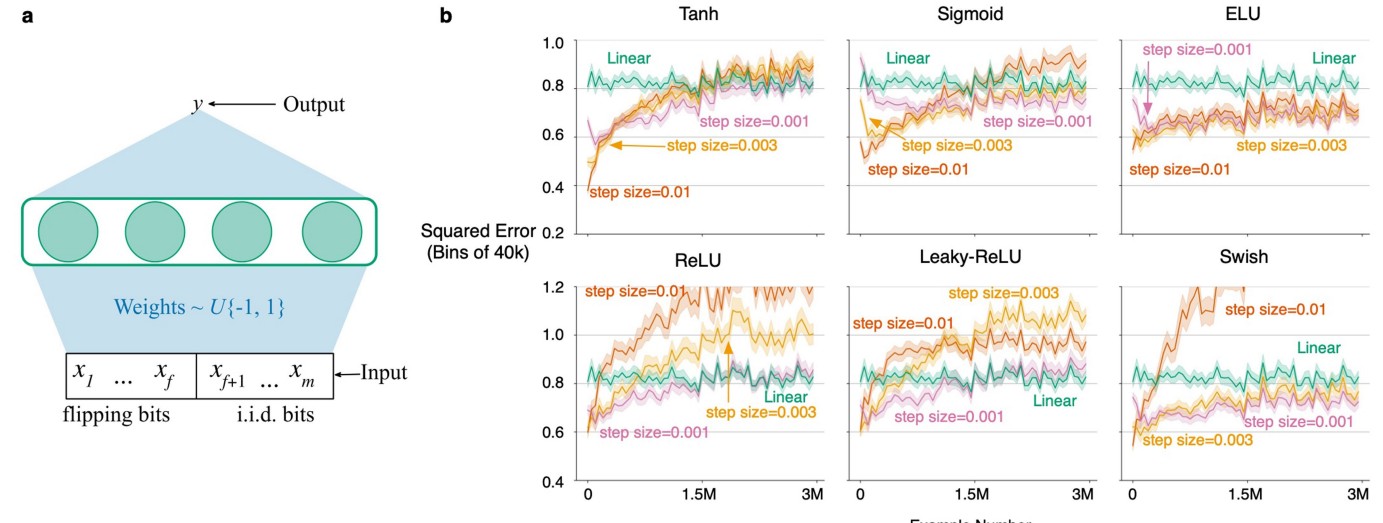

**Extended Data Fig. 2 | Loss of plasticity in the Slowly-Changing Regression problem. a**, The target function and the input in the Slowly-Changing Regression problem. The input has $m + 1$ bits. One of the flipping bits is chosen after every $T$ time steps and its value is flipped. The next $m - f$ bits are i.i.d. at every time step and the last bit is always one. The target function is represented by a neural network with a single hidden layer of LTUs. Each weight in the target network is −1 or 1. **b**, Loss of plasticity is robust across different activations. These results are averaged over 100 runs; the solid lines represent the mean and the shaded regions correspond to ±1 standard error.

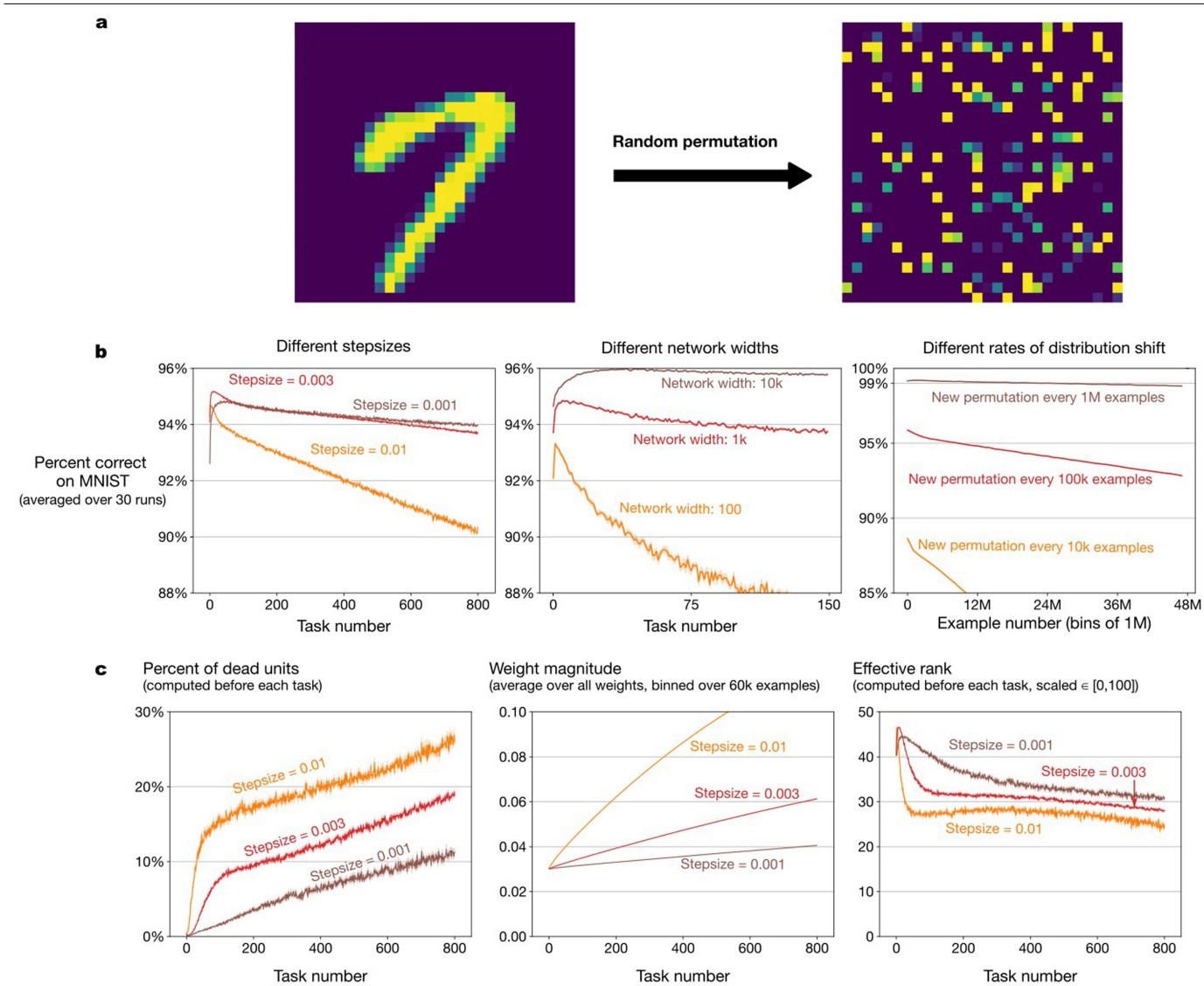

**Extended Data Fig. 3 | Loss of plasticity in Online Permuted MNIST. a**, Left, an MNIST image with the label '7'; right, a corresponding permuted image. **b**, Loss of plasticity in Online Permuted MNIST is robust over step sizes, network sizes and rates of change. **c**, Evolution of various qualities of a deep network trained by means of backpropagation with different step sizes. Left, over time, the percentage of dead units in the network increases. Centre, the average magnitude of the weights increases over time. Right, the effective rank of the representation of the networks trained with backpropagation decreases over time. The results in these six plots are the average over 30 runs. The solid lines represent the mean and the shaded regions correspond to ±1 standard error. For some lines, the shaded region is thinner than the line width, as standard error is small.

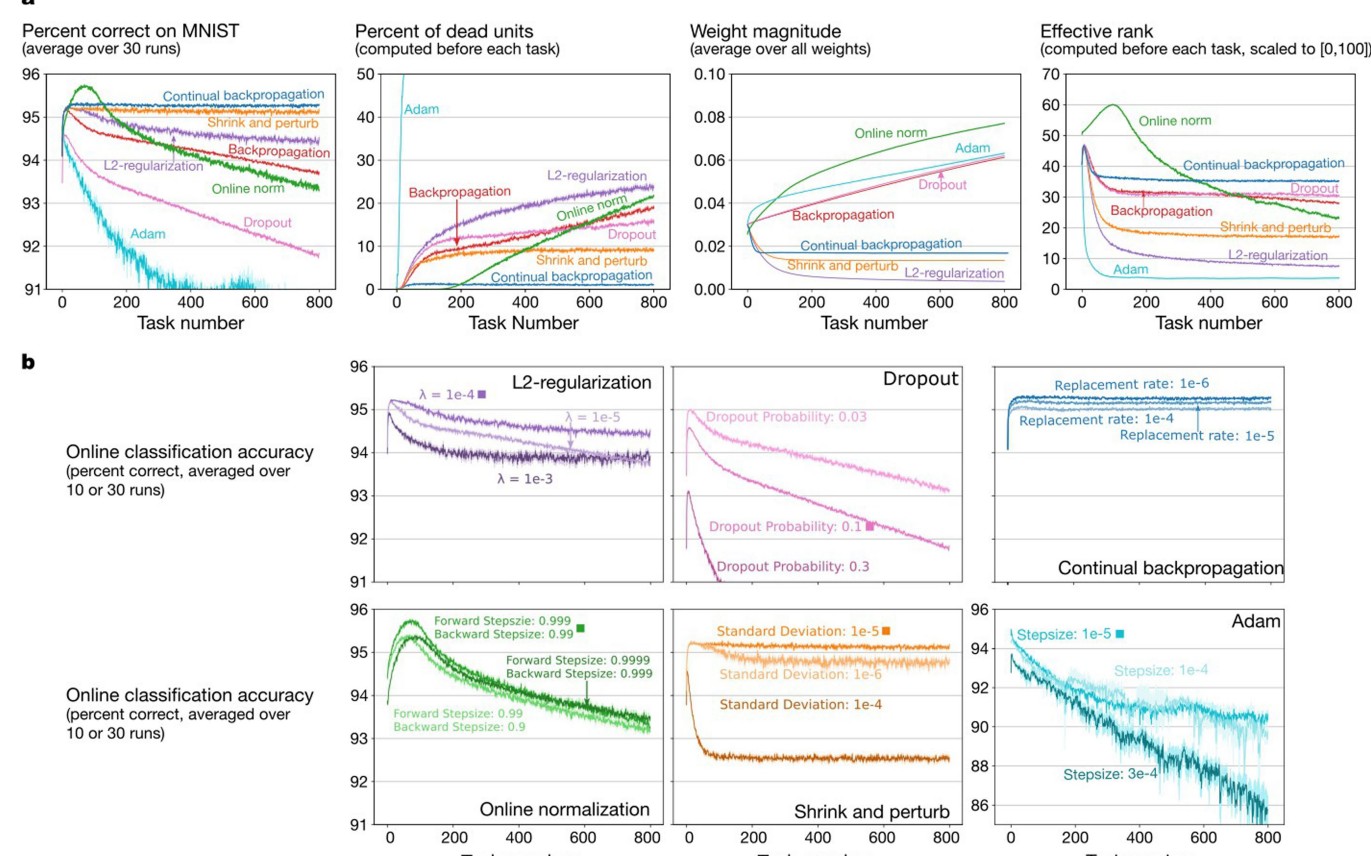

**Extended Data Fig. 4 | Existing deep-learning methods on Online Permuted MNIST. a**, Left, online classification accuracy of various algorithms on Online Permuted MNIST. Shrink and Perturb has almost no drop in online classification accuracy over time. Continual backpropagation did not show any loss of plasticity and had the best level of performance. Centre left, over time, the percentage of dead units increases in all methods except for continual backpropagation; it has almost zero dead units throughout learning. Centre right, the average magnitude of the weights increases over time for all methods except for L2 regularization, Shrink and Perturb and continual backpropagation. These are also the three best-performing methods, which suggests that small weights are important for fast learning. Right, the effective rank of the representation of all methods drops over time. However, continual backpropagation maintains a higher effective rank

than both backpropagation and Shrink and Perturb. Among all the algorithms, only continual backpropagation maintains a high effective rank, low weight magnitude and low percentage of dead units. The results correspond to the average over 30 independent runs. The shaded regions correspond to ±1 standard error. **b**, Performance of various algorithms on Online Permuted MNIST for various hyperparameter combinations. For each method, we show three different hyperparameter settings. The parameter settings that were used in the left panel in **a** are marked with a solid square next to their label. The results correspond to the average of over 30 runs for settings marked with a solid square and 10 runs for the rest. The solid lines represent the mean and the shaded regions correspond to ±1 standard error.

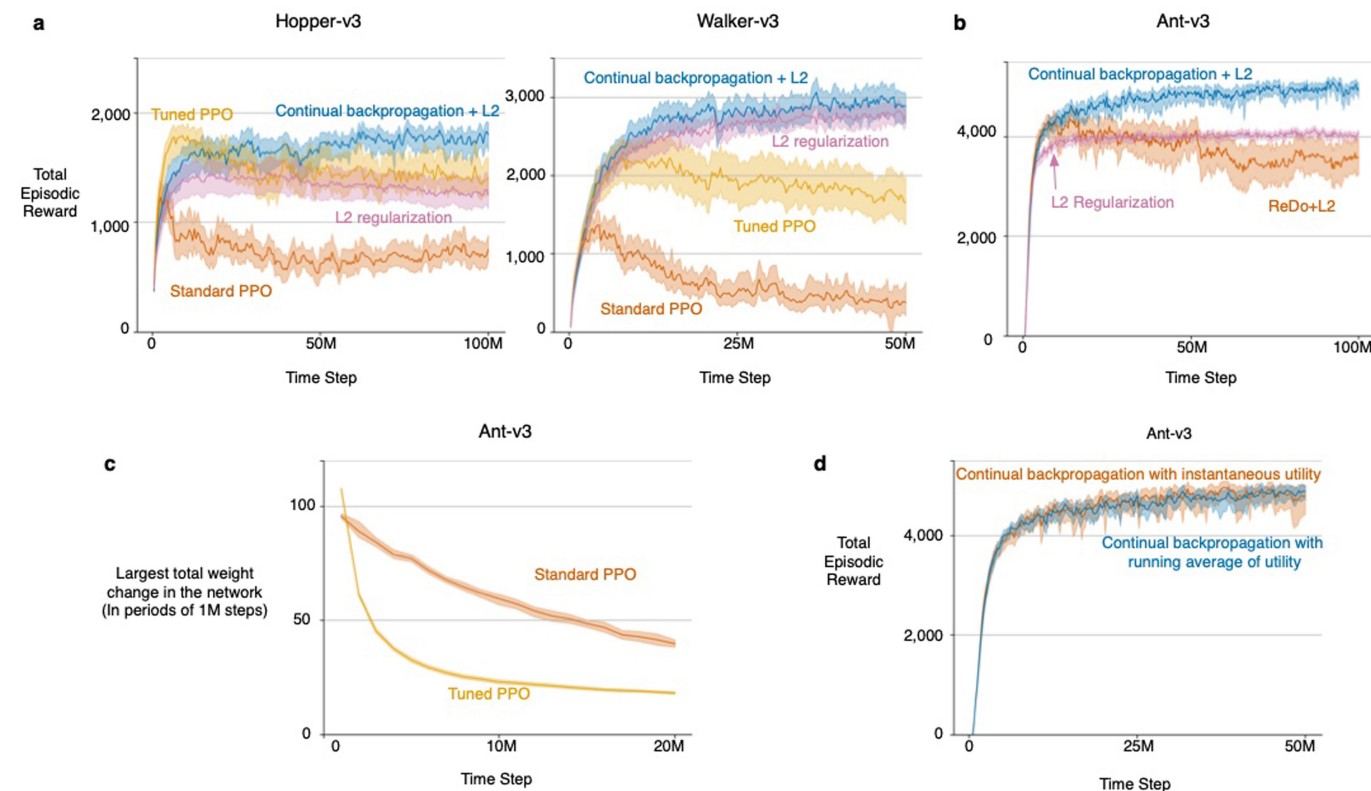

**Extended Data Fig. 5 | Further results in stationary reinforcement-learning problems. a**, Similar to Fig. 4, the performance of standard PPO drops over time. However, unlike in Fig. 4, the performance of PPO with L2 regularization gets worse over time in Hopper-v3. On the other hand, PPO with continual backpropagation and L2 regularization can keep improving with time. **b**, Comparison of continual backpropagation and ReDo on Ant-v3. The performance of PPO with ReDo and L2 regularization worsens over time, whereas PPO with continual backpropagation and L2 regularization keeps improving over time. **c**, PPO with standard Adam leads to large updates in the policy network compared with proper Adam ($\beta_1 = \beta_1 = 0.99$), which explains why PPO with proper Adam performs much better than standard PPO. **d**, Comparison of two forms of utility in continual backpropagation, when using a running estimate of instantaneous utility and when using just the instantaneous utility. Both variations have similar performance. All these results are averaged over 30 runs; the solid lines represent the mean and the shaded regions correspond to 95% bootstrapped confidence interval.

**Extended Data Table 1 | Details of the ResNet-18 architecture used for the class-incremental CIFAR-100 problem**

| Layer Name | Layer Type | Layer Parameters | Receives Inputs From |
|---|---|---|---|
| Input Layer | - | Input Shape: (3, 32, 32) | - |
| Conv 1 | Conv + Batch Norm + ReLU | Out Filters: 64, Stride: 1 | Input Layer |
| Conv 2 | Conv + Batch Norm + ReLU | Out Filters: 64, Stride: 1 | Conv 1 |
| Conv 3 | Conv + Batch Norm | Out Filters: 64, Stride: 1 | Conv 2 |
| Act 3 | ReLU Activation | - | Conv 3 + Conv 1 |
| Conv 4 | Conv + Batch Norm + ReLU | Out Filters: 64, Stride: 1 | Act 3 |
| Conv 5 | Conv + Batch Norm | Out Filters: 64, Stride: 1 | Conv 4 |
| Act 5 | ReLU Activation | - | Conv 5 + Act 3 |
| Downsample 6 | Conv + Batch Norm | Out Filters: 128, Stride: 2 | Act 5 |
| Conv 6 | Conv + Batch Norm + ReLU | Out Filters: 128, Stride: 2 | Act 5 |
| Conv 7 | Conv + Batch Norm | Out Filters: 128, Stride 1 | Conv 6 |
| Act 7 | ReLU Activation | - | Conv 7 + Downsample 6 |
| Conv 8 | Conv + Batch Norm + ReLU | Out Filters: 128, Stride 1 | Act 7 |
| Conv 9 | Conv + Batch Norm | Out Filters: 128, Stride 1 | Conv 8 |
| Act 9 | ReLU Activation | - | Conv 9 + Act 7 |
| Downsample 10 | Conv + Batch Norm | Out Filters: 256, Stride: 2 | Act 9 |
| Conv 10 | Conv + Batch Norm + ReLU | Out Filters: 256, Stride: 2 | Act 9 |
| Conv 11 | Conv + Batch Norm | Out Filters: 256, Stride: 1 | Conv 10 |
| Act 11 | ReLU Activation | - | Conv 11 + Downsample 10 |
| Conv 12 | Conv + Batch Norm + ReLU | Out Filters: 256, Stride: 1 | Act 11 |
| Conv 13 | Conv + Batch Norm | Out Filters: 256, Stride: 1 | Conv 12 |
| Act 13 | ReLU Activation | - | Conv 13 + Act 11 |
| Downsample 14 | Conv + Batch Norm | Out Filters: 512, Stride: 2 | Act 13 |
| Conv 14 | Conv + Batch Norm + ReLU | Out Filters: 512, Stride: 2 | Act 13 |
| Conv 15 | Conv + Batch Norm | Out Filters: 512, Stride: 1 | Conv 14 |
| Act 15 | ReLU Activation | - | Conv 15 + Downsample 14 |
| Conv 16 | Conv + Batch Norm + ReLU | Out Filters: 512, Stride: 1 | Act 15 |
| Conv 17 | Conv + Batch Norm | Out Filters: 512, Stride: 1 | Conv 16 |
| Act 17 | ReLU Activation | - | Conv 17 + Act 15 |
| Pooling Layer | Average Pooling + Flatten Layer | Output Size: 512 | Act 17 |
| Output Layer | Linear | Output Size: 10 | Pooling Layer |

All conv layers used a kernel size of (3, 3), reshape layers used a kernel size of (1, 1) and the pool layer used a kernel size of (4, 4).

**Extended Data Table 2 | Hyperparameter selection in Continual ImageNet**

| Algorithm Name | Stepsize | Weight Decay / Replacement Rate | Noise Variance |
|---|---|---|---|
| **L2-Regularization** | 0.1, **0.03**, 0.01, 0.003 | $3*10^{-5}$, $10^{-5}$, $3*10^{-6}$, $10^{-6}$ | - |
| **Shrink and Perturb** | 0.03, **0.01** | $3*10^{-5}$, $10^{-5}$, $3*10^{-6}$, $10^{-6}$ | $10^{-4}$, $10^{-5}$, $10^{-6}$, $10^{-7}$ |
| **Continual Backpropagation** | 0.03, **0.01** | $3*10^{-3}$, $10^{-3}$, $3*10^{-4}$, $1*10^{-4}$, $3*10^{-5}$ | - |

Values used for the grid searches to find the best set of hyperparameters for all algorithms tested on Continual ImageNet. The best-performing set of values for each algorithm is in bold. The values in the third column for L2 regularization and Shrink and Perturb correspond to the weight decay, whereas for continual backpropagation, they correspond to the replacement rate.

**Extended Data Table 3 | Details of the artificial neural network used for the Continual ImageNet problem**

| Layer 1: Convolutional + Max-Pooling | | | | |
|---|---|---|---|---|
| Number of Filters | 32 | Activation | | ReLU |
| Convolutional Filter Shape | (5,5) | Convolutional Filter Stride | (1,1) | |
| Max-Pooling Filter Shape | (2,2) | Max-Pooling Filter Stride | | (1,1) |

| Layer 2: Convolutional + Max-Pooling | | | |
|---|---|---|---|
| Number of Filters | 64 | Activation | ReLU |
| Convolutional Filter Shape | (3,3) | Convolutional Filter Stride | (1,1) |
| Max-Pooling Filter Shape | (2,2) | Max-Pooling Filter Stride | (1,1) |

| Layer 3: Convolutional + Max-Pooling | | | |
|---|---|---|---|
| Number of Filters | 128 | Activation | ReLU |
| Convolutional Filter Shape | (3,3) | Convolutional Filter Stride | (1,1) |
| Max-Pooling Filter Shape | (2,2) | Max-Pooling Filter Stride | (1,1) |

| Layer 4: Fully Connected | | | |
|---|---|---|---|
| Output Size | 128 | Activation | ReLU |

| Layer 5: Fully Connected | | | |
|---|---|---|---|
| Output Size | 128 | Activation | ReLU |

| Layer 6: Fully Connected | | | |
|---|---|---|---|
| Output Size | 2 | Activation | Linear |

The network has three convolutional layers followed by three fully connected layers.

**Extended Data Table 4 | Implementation details for the Slowly-Changing Regression problem and the learning network**

## Slowly-Changing Regression Problem Parameters

| Parameter Name | Description | Value |
|---|---|---|
| m | Number of input bits | 21 |
| f | Number of flipping bits | 15 |
| n | Number of hidden units | 100 |
| T | Duration between bit flips | 10,000 steps |
| Bias | Included in input and output layers | True |
| $\theta_i$ | LTU Threshold | $(m+1)\cdot\beta - S_i$ |
| $\beta$ | Proportion used in LTU Threshold | 0.7 |

## Learning Network Parameters

| Parameter Name | Value |
|---|---|
| **Number of hidden layers** | 1 |
| **Number of units in each hidden layer** | 5 |

The target and the learning networks both have a single hidden layer.

**Extended Data Table 5 | Hyperparameters for PPO**

| Name | Default Value |
|---|---|
| Policy network | (256, ReLU, 256, ReLU, Linear) + Standard deviation variable |
| Value network | (256, ReLU, 256, ReLU, Linear) |
| Buffer size | 2048 |
| Number of epochs | 10 |
| Mini-batch size | 128 |
| GAE, $\lambda$ | 0.95 |
| Discount factor, $\gamma$ | 0.99 |
| Clip parameter | 0.2 |
| Input normalization | False |
| Advantage normalization | True |
| Value function loss clipping | False |
| Gradient clipping | False |
| Optimizer | Adam |
| Optimizer step size | $10^{-4}$ |
| Optimizer $\varepsilon$ | $10^{-8}$ |

All reinforcement-learning algorithms share these hyperparameters. Additional hyperparameters are described in the 'Details and further analysis in reinforcement learning' section in the Methods.