## [Peer Review File · Nature]

Manuscript Title: Loss of Plasticity in Deep Continual Learning

Reviewer Comments & Author Rebuttals

Reviewer Reports on the Initial Version:

Referees' comments:

Referee #1 (Remarks to the Author):

A (Summary):

This paper explores the loss of plasticity in continual learning, and demonstrate that most of the deep learning methods are unable to maintain plasticity for continual learning tasks.

Through a series of supervised learning tasks, modified to be continual learning problems, the authors empirically demonstrate the loss in plasticity occurring.

The authors propose a new algorithm, continual backpropagation, that addresses most of the issues encountered in the above analysis, and demonstrate that alternative techniques are unable to address all of the issues observed.

B (Originality):

Although this work builds on existing ideas (dead neurons, gradient/weight-based scoring), their combination is novel. The main issue I see in terms of originality is with its close resemblance to [1] (but more on that below).

C (Data & methodology):

The experiments conducted were reasonably thorough, although the tasks are quite small. I can appreciate that the breadth of experiments run make it difficult to run on more challenging environments, given limited resources.

My main concern on data used is that all of the tasks considered are quite artificial: they are modifications of standard supervised learning methods, but it is not clear how reflective they are of real-world continuous problems. I understand that there may not be standard benchmarks for continual learning, but it would have been better to focus a bit more on existing problems, rather than customized ones. Indeed, as mentioned by the authors in the paper, reinforcement learning is a problem that can be considered as a continual learning problem, yet the authors still chose to modify an existing environment. The authors mention replay buffers and target networks as mechanisms the RL community has used to become "train-once" tasks, but it seems more natural to me to use RL tasks without a replay buffer and target network, rather than modifying the environment dynamics.

That being said, focusing on RL would necessitate comparison with [1], which I discuss further below.

D (Appropriate use of statistics and treatment of uncertainties):

The statistics and uncertainties reported are satisfactory to me.

E (Conclusions: Robustness, validity, reliability):

The authors are convincing in demonstrating that existing methods from deep learning are unable to address plasticity loss in continual learning tasks. Although not terribly surprising, as these methods were not developed for continual learning problems, it is still a valid finding.

I am a bit less convinced about the robustness of their proposed method (continual backprop). It seems to be quite expensive to run (but neither runtime nor walltime are reported, yet it seems like the advantages over shrink and perturb and the "Mean-corrected contribution" ablation are rather marginal. For instance:

1. In Figure 4a, there is only a very mild degradation for backprop; additionally, the percentage correct for all. are in the range of 91-96, which is still pretty high. This makes me wonder if this is too easy a problem to evaluate on? For this particular analysis, it would be good to get at least some sense of whether the results are similar in more difficult problems such as the ImageNet one.
2. In line 474 you say "None of the existing methods fully maintain plasticity", but shrink and perturb does seem to be able to do so. Further you later say both "S&P and L2-regularization are very sensitive to hparam values", but it doesn't really appear to be the case in Figure 1 in the Appendix.

F (Suggested improvements):

Some suggestions for improvements, in order of appearance:

1. You mention replay buffers, batching, and target networks make RL "very nearly a train-once setting", but it's not entirely clear why, so it would be good to elaborate this a bit more. For instance, if you add a replay buffer to one of the continual learning tasks you consider, is it no longer a continual learning task?
2. Continuing the above point, it's not clear where RL lies in your definition of continual learning. At one point you say "whenever the house's layout changed... the robot could simply learn from the new data", but you're conflating changing data versus changing environments, which are two different things. A more general term, that you do use further down, is "changes in their data streams". I would suggest you adopt this one everywhere.
3. You say "A classical deep learning problem like ImageNet is the best place to show the issue of loss of plasticity", but this is not necessarily true. You're using a customized version of ImageNet, not the original benchmark.
4. In lines 149-151, couldn't you just output one number, corresponding to the probability of belonging to one of the classes, instead of two heads?
5. If adding new heads for new classes is the standard in these types of problems, then I don't think resetting heads, as you do, is equivalent.
6. In line 215-216, if you did not use convolutional layers, does this mean the images were flattened? In what order are the pixels presented in this case?
7. In Figure 3, it would be good to add the values in the non-continual learning setting as reference.
8. In Figure 3, right panel, what is the maximum possible value?
9. In line 378 when you say "improving the condition number", does this mean increasing or decreasing the ratio?
10. In line 412 you say "we only present results for one set of these hyperparameters". But in the appendix you say "for all the methods except for dropout, we selected the hyperparameter setting that resulted in the highest average percent correct during the whole training period". Why was

dropout handled differently?

11. In lines 456-462, is this just poor hparam choice? In Appendix A Adam with stepsize $1e-5$ had better performance. Why didn't you use that? Further, your discussion in lines 466-468 go against the strong results we see with deep RL and Adam, which suggest that the findings might be more due to poor hparam choice.

12. In line 509 when you say "a new hidden unit is added", do you mean reinitialized?

13. Instead of using a replacement-rate, you could also use a threshold on utility instead (similar to [1]). It would be nice if this were evaluated.

14. The math in figure 5 could be made clearer.

15. I appreciate the motivation of your utility measure, but it is quite complicated and it's not clear it needs to be. As mentioned elsewhere in this review, your results with "Mean-corrected contribution" in your ablations, as well as the comparison done in [1] suggest that the utility measure can be much simpler. Perhaps you do need it for your particular setting, but it would require a comparison with ReDo in order to make it convincing.

16. On a similar note: it seems very expensive to do the reinitialization at each update. Why not do this less frequently, as was done in [1], for instance?

17. Given the strong connection between reinforcement learning and continual learning, why not include the reinforcement learning results in the main paper? They are somewhat hidden in the appendix and there is only a rather small discussion in the main paper.

18. In line 670 you say "the benefits of injected noise are controversial because it can be tricky to inject noise to avoid poor performance". Why not try them in your settings? They seem like natural baselines.

G (References): This is one of the weakest points in this paper, as the authors are missing proper discussions of some papers directly related to their work. In particular, [1] is exploring a very similar issue (but named dormant neurons) and propose a simpler algorithm for addressing them. In fact, [1] does compare against continual backprop, based on the ArXiv version of this submission, in Appendix C.4. Their results suggest there is no real gain from the scoring method used here, so this requires much further discussion.

Some places where I think more comparisons with this paper are necessary are:

1. Lines 288-293
2. Need to compare to ReDo from [1] in section 5

You do cite [1] in the very last paragraph of your submission, and only listed amongst other papers. Given that this is the most related paper, relegating it to the end in such a shallow manner is certainly not enough for publication.

Some other works that should be cited

1. [2], in particular in line 389 when discussing Adam in deep RL.
2. In "line" 521 you only mention one type of pruning, but there are many others (see [3] for a good survey).
3. In line 623 you should also cite [4].
4. In line 630 you should also cite [5].

5. In the last paragraph in page 26 you should also be citing [1], [6],

H (Clarity and context):

The writing is relatively clear, although it could benefit from an editorial pass. There are a few typos scattered throughout, and many sentences start with "And ...", which although not incorrect, reads a little awkwardly.

Also, all of the math equations seem to be garbled, so I was unable to verify them.

[1] Sokar, G., Agarwal, R., Castro, P.S. & Evci, U.. (2023). The Dormant Neuron Phenomenon in Deep Reinforcement Learning. *Proceedings of the 40th International Conference on Machine Learning*, in *Proceedings of Machine Learning Research* 202:32145-32168 Available from <https://proceedings.mlr.press/v202/sokar23a.html>.

[2] Ceron, J.S.O. & Castro, P.S.. (2021). Revisiting Rainbow: Promoting more insightful and inclusive deep reinforcement learning research. *Proceedings of the 38th International Conference on Machine Learning*, in *Proceedings of Machine Learning Research* 139:1373-1383 Available from <https://proceedings.mlr.press/v139/ceron21a.html>.

[3] Gale, T., Elsen, E., and Hooker, S. The state of sparsity in deep neural networks. <https://arxiv.org/abs/1902.09574>

[4] Evci, U., Gale, T., Menick, J., Castro, P.S. & Elsen, E.. (2020). Rigging the Lottery: Making All Tickets Winners. *Proceedings of the 37th International Conference on Machine Learning*, in *Proceedings of Machine Learning Research* 119:2943-2952 Available from <https://proceedings.mlr.press/v119/evci20a.html>.

[5] Graesser, L., Evci, U., Elsen, E. & Castro, P.S.. (2022). The State of Sparse Training in Deep Reinforcement Learning. *Proceedings of the 39th International Conference on Machine Learning*, in *Proceedings of Machine Learning Research* 162:7766-7792 Available from <https://proceedings.mlr.press/v162/graesser22a.html>.

[6] Pierluca D'Oro, Max Schwarzer, Evgenii Nikishin, Pierre-Luc Bacon, Marc G Bellemare, Aaron Courville. Sample-Efficient Reinforcement Learning by Breaking the Replay Ratio Barrier. ICLR 2023. <https://openreview.net/forum?id=OpC-9aBBVJe>

[7] Schwarzer, M.*, Obando Ceron, J.S.*, Courville, A., Bellemare, M.G., Agarwal, R. & Castro, P.S.. (2023). Bigger, Better, Faster: Human-level Atari with human-level efficiency. *Proceedings of the 40th International Conference on Machine Learning*, in *Proceedings of Machine Learning Research* 202:30365-30380 Available from <https://proceedings.mlr.press/v202/schwarzer23a.html>.

Referee #2 (Remarks to the Author):

This manuscript provides a neat demonstration that deep neural networks sequentially trained on many tasks can suffer from substantial loss of plasticity. While, as the authors themselves discuss in the manuscript, this observation is not completely novel or unexpected, I agree with the authors' claim that they provide the clearest and most direct demonstration so far. Moreover, loss of plasticity is a rather under-appreciated phenomenon in both the reinforcement learning and the continual learning literature, and this manuscript does a great job highlighting this phenomenon and arguing why it is important. In this regard it also greatly helps that the article is skilfully written and accessible to a broad audience. In short, I think that the part of this manuscript that demonstrates the problem of plasticity loss in deep continual learning is generally of high quality and likely to be impactful. The analysis aimed at understanding the factors underlying loss of plasticity is also interesting and nicely done.

However, I was rather disappointed by the solution – continual backpropagation – that is proposed in the manuscript, as well as by the way this solution is evaluated. It seems obvious that periodically reinitialising part of the network should help mitigate loss of plasticity. It also does not seem very striking or interesting that such periodic reinitialising does this more effectively than methods with different motivations such as L2-regularisation, dropout or online normalisation. Instead, if this proposed solution is included, I would expect the authors to address questions such as whether periodically reinitialising part of the network can be done without catastrophic forgetting (i.e., whether this approach strikes a good balance in the plasticity/stability dilemma, e.g. better than current approaches for continual learning), or what the benefit is of only partly reinitialising the network compared to fully reinitialising it. Perhaps an option could also be to leave out the part about the proposed solution, or to include it in minimal form with a clear discussion of its limitations?

My relevant expertise with regards to this manuscript is mostly in continual learning. As a brief summary of my review, I consider the provided insights regarding plasticity loss to be of high value to the continual learning field, these insights will be useful for developing better approaches for continual learning; but I am not convinced that the proposed selective reinitialisation is such an approach.

List of issues:

{1} Regarding the first part of the manuscript on the description of the problem I have one comment, and that is regarding the claim that loss of plasticity is “more fundamental” than catastrophic forgetting (e.g., in the abstract and in the introduction). This is a strong claim, but I could not find a justification for it in the manuscript. I also think that this claim is too strong, and risks raising too high expectations. Let's take reinforcement learning as an example. The issue of forgetting is well-recognized in this field, and addressing it by means of experience replay has resulted in major improvements. Do the authors want to imply that by addressing the issue of loss of plasticity, even larger improvements can be expected?

{2} The manuscript is rather naive with respect to the existing continual learning literature, especially

when it comes to the evaluation of the proposed continual backpropagation. It is generally agreed in the literature that the goal of continual learning is developing models that can incrementally learn new skills with minimal forgetting of past skills, but this manuscript simply ignores forgetting, and does not evaluate the performance of the continually trained models on previously learned tasks. The justification for this seems to be that the focus of this manuscript is on loss of plasticity, and that this is complementary to the already more-common focus on catastrophic forgetting. While I agree that a *focus* on loss of plasticity is original and commendable (this perspective could help to develop better methods for continual learning), this does not mean that it is interesting or justified to then only evaluate plasticity in isolation from forgetting. This is like studying the plasticity/stability dilemma by only considering plasticity: if stability is disregarded, of course maximising plasticity becomes a lot easier, but also much less interesting. (Note that it is also not the case that current continual learning studies only evaluate forgetting: the main metric in continual learning is average final performance over all tasks, which includes both forgetting and plasticity.)

{3} The authors should discuss what is the benefit of only partially re-initialising the network compared to the naive solution of fully re-initialising the network for each new task. I also think that this solution of complete re-initialisation for each new task should be included in the empirical comparisons. For the experiments on ImageNet, my impression is that this naive baseline would be very competitive with continual backpropagation (at least when only considering the performance on the current task). From Figure 1 we can see that the performance for a (re-)initialised network trained on a single task is near 90% (and this could be higher if the step size is tuned for maximal performance after a single task). If a network would be fully re-initialised for each new task, this same performance would be expected for each task in the task sequence. Comparing this with Figure 7, this means that for most tasks in the task sequence the performance of the naive baseline of full re-initialisation would be higher than the performance of continual backpropagation.

{4} The claim that “the standard practice in deep continual learning [...] is to introduce new heads for new classes” (L161-162, p6) is not quite true. It would be fair to claim that introducing new heads is the standard practice for task-incremental learning, but for other types of continual learning such as domain- and class-incremental learning this is not the case.

Minor things:

- L65 on p24: last mention of supervised should be reinforcement learning?
- L744 on p27: “optimizes” -> “optimizers”

(Line numbers refer to the outer set of line numbers.)

As a final comment, the layout of the submitted manuscript was distorted at several places (e.g., formulas and some figures). For the reviewing I therefore mostly based myself on the latest arXiv version of this manuscript (<https://arxiv.org/abs/2306.13812v2>), which based on spot checks seemed similar to the submitted manuscript except with better layout.

Referee #3 (Remarks to the Author):

The paper discusses loss of plasticity, a particular phenomenon that happens when learning in non-stationary settings, and proposes continual backprop as a solution to this problem. Loss of plasticity is specific to neural networks, and it manifests by the network not being able to learn new information. The authors argue that one main culprit behind this is saturating units, alongside growth of weight norm and collapse of the representation rank. Continual backprop addresses this problem by resetting units based on their lack of utility to the model, hence allowing them to be recycled by the learning process.

The work is original. In particular the authors are the first to start the discussion around this form of loss of plasticity (through an arxiv paper), and the work spurred a few follow-ups, particularly in the RL community, and therefore it already had an impact.

That stated, I have a few questions and comments that I feel the authors should address:

I'm not sure I agree with the "train-once" terminology. Traditionally people will refer to this as either the stationary setting or the IID setting. I'm not sure why we need a new term. In my opinion this just creates confusion. Is the train-once meant to convey something different from saying the IID setting / stationary setting? I'm also worried about confusion with one-epoch strategies, where each data point can only be seen once, which have become more popular lately.

Ash & Adams original work on warm-starting, which is also the starting point of Berariu et al., explores a (slightly) different phenomenon than the one described in this manuscript. Unfortunately both are referred to as loss of plasticity, though Berariu et al. does highlight this discrepancy. In these two works loss of plasticity is defined as inability to generalize. Namely, training error can be driven as low as a freshly initialized model, while validation error stays high (or ends up being higher than the freshly initialized model). In contrast, the loss of plasticity studied here has more to do with the optimization process, because the learning objective can not be optimized. Saturating units lead to no gradient being propagated for example, and hence optimization fails. Recent works (e.g. Lyle et al. 2023), also acknowledge this difference.

I urge the authors to discuss this point in the paper. The use of the same term is unfortunate, I'm not advocating that this paper has to correct the issue. But in this work the authors claim that the first observation of loss of plasticity is Ash and Adams paper, clearly implying that it is the same problem. I strongly disagree that it is the same phenomenon. So I'd like the authors to motivate properly why it might be the same phenomenon if that is the stance they would like to take, or to acknowledge that these works might be studying something different.

I think the authors should better justify the choice of architecture for ImageNet. I can see the architecture as being meaningful potentially as it is very popular in RL (it is basically a variant of DQN). But is very far from what would be standard practice in computer vision, particularly for

ImageNet. The fact that you do binary classification does not justify departing from more established architecture designed choices that include residual connections, like resnet. I think it is ok not to run resnet because the target might be RL. But for any computer vision person the choice is unnatural, and acknowledging this point would be useful.

All datasets provided are synthetic. In particular permuted MNIST is generally seen as being too toyish in the continual learning community and people are dissuaded from using it. I can understand that here the authors go through 150 tasks (vs typical Continual learning papers that would do 10). But an issue with permuted MNIST is not scale but the fact that there is no structure to share between the tasks. It would be interesting to know what happens when there is common structure between tasks as one would expect in more realistic scenarios (ImageNet represents this scenario). In that sense I would suggest to bring the ant task from the appendix into the main text, since it is in my view more interesting than the permuted MNIST.

Comment on dropout. Since it encourages redundant representation (i.e. same information being encoded in multiple features) wouldn't one expect to reduce the rank and hence not help?

I would have preferred a more technical explanation of how each method should help. E.g. for Adam, why should it help with non-stationarity? The algorithm was developed for the IID setting and aims to make better use of curvature information. Has anyone connected its use in RL with non-stationarity? I could see it as just being a better optimizer but as affected by non-stationarity as SGD. I agree that it is important to run experiments with different optimizers, particularly widely used ones like Adam. But the motivation is different. Not that they should help, but rather just to make sure the results hold in typical scenarios. From how the text is written, it implies that there are some reasons we should expect Adam to help improve plasticity (which is actually not spelled out or explored).

The authors identify 3 causes for loss of plasticity: dead units, weight magnitude and low rank. The authors do not fully go into explaining why these 3 affect plasticity. E.g. for dead units the paper states that later on (next section) will show that if we reduce the number of dead units we have more plasticity, implying their causal role in plasticity loss. Fair enough, but it would also help to explain how they are connected. E.g. say something to the effect that dead units do not allow gradients to flow, hence you can not change weights going in the dead unit. For weight magnitude authors mention exploding gradients in RNNs, which is potentially irrelevant for the work since there are just shallow feedforward networks. A more relevant approach was to connect this to curvature and potentially ill-conditioning of the optimizer. Basically rely on traditional tools to understand learning and being more precise. The gradients being bounded argument also doesn't hold, because they are technically bounded when learning fails (probably larger than those on the early tasks, but bounded). Right now there is very little precise information of why these 3 metrics matter and no discussion whether there are other properties that matter. And the authors do not really provide a technical description of how these three items interfere with the learning process, leaving things in my opinion a bit too vague.

In the same spirit of the comment above. When introducing the utility measure, could the authors provide more precise technical reasons of why the metric makes sense? E.g. the authors mention

sparsity using a similar metric. Usually in sparsity the metric is motivated using Taylor expansions that gives you a mathematical tool to say how much zeroing out a weight will affect the output of the function. Similarly, adaptability maybe can be connected with how much the function can change under an update step for some metric? None of this is done in the paper. The explanation is intuitive, but not very technical or precise.

Subtracting the average of utility might imply a Gaussian assumption on the activation, right? Such assumptions should be made explicit. What other assumptions are made in terms of how activations are meant to work?

Connected to a previous point about permuted MNIST, but I think important for the paper is a discussion on the tradeoffs between maintaining ability to learn, being able to learn to reuse previous knowledge, performance that the model achieves on the given task, maybe even catastrophic forgetting. In particular I find that the authors do not integrate the work into the different concerns that people have in the community. I would appreciate a paragraph describing these different phenomena that have been studied in the community. For example, a considerable section of the community is focusing on the concept of forward transfer. The goal is not only to be able to keep learning, but also to learn more efficiently as time goes by because you get to reuse skills and knowledge that you have accumulated so far. From this perspective, fixing the optimization to have the neural network able to still learn feels like a first step but not sufficient. If future data becomes similar (but not the same) with previous data, then resetting certain units that seem to have low utility might be harmful. These considerations affect the choice of benchmark. For example, in permuted MNIST there is no shared structure, so questions like whether continual backprop learns as fast (or faster) then learning from scratch can not be asked in a meaningful way. ImageNet is not potentially set up in such a way to see whether one can find itself in a situation where locally a unit might seem not useful, but globally it will be. I'm not suggesting that the authors need to address or experiment with any of these alternatives. But I think acknowledging the existence of these themes, and how the choice of benchmark affect these questions will give to the reader a much better and rounded perspective of the topics explored in the community and the role of loss of plasticity within this context.

Author Rebuttals to Initial Comments:

Response from the authors

We want to start by thanking all the referees and the editor for their detailed and insightful comments, which have led us to substantially improve the article. We are glad to see that all the referees and the editor appreciate our demonstrations of loss of plasticity and recognize the importance of this issue for both the continual and reinforcement learning communities.

We have made many changes to the article to address all the concerns raised by the referees and the editor. We have also reduced the size of the main paper to be suitable for publication in Nature. Here are the main five changes.

1. **Addition of ResNet results:** The main limitation of our supervised learning experiments was that we did not use architectures that are used in practice. In our new experiments in a class-incremental setting using the CIFAR-100 dataset, we show that even popular architectures lose plasticity and that continual backpropagation completely fixes the issue. Additionally, the class-incremental problem is relevant in practice as it represents the typical scenario when the learning system keeps getting more and more data and has to accumulate new knowledge.
2. **Addition of RL experiments:** The second set of new results are in stationary reinforcement learning problems. We used standard Mujoco environments, using the PPO algorithm with standard hyper-parameters, and found that the performance of PPO drops to a very low level when run longer than what is done in the literature. The inability of PPO to relearn a good policy after the performance drops is another demonstration of loss of plasticity. Additionally, we show that continual backpropagation fully overcomes this drop in performance. These results in standard reinforcement learning environments with a popular algorithm like PPO make plasticity loss much more relevant to reinforcement learning applications.
3. **Changes to continual backpropagation:** We found that the utility measure we used in the previous manuscript was only necessary for activation functions that saturate at a non-zero value (tanh, ELUs, etc.). For activations like ReLU, the contribution utility works almost as well as the overall utility from the original manuscript. So, we decided only to use continual backpropagation with contribution utility. This simplifies the algorithm, making it much easier to adopt in practice.
4. **Restructuring the article.** We have reduced the main body's size to about 4000 words to make it small enough for publication. To reduce the size of the main article,

we moved all the MNIST experiments, a big part of the discussion, slowly changing regression and all of the details to Methods. Additionally, we brought the Slippery Ant experiments to the main paper. Still, the methods section needed to be shorter, and we had to cut down many parts of the writing and explanations to make Methods shorter.

5. **Changes to the slippery ant experiment:** In the previous experiment, we used a very wide range for the coefficient of friction. We found that range unrealistic in practice and have now changed the friction coefficient to between 0.02 and 2.

We look forward to further discussion with the referees to further refine the article. We now provide a detailed response to all the referees.

Response to the Editor

- Refs acknowledge the thoroughness of the demonstration of the problem using permuted MNIST tasks and imagenet, but raise that this isn't a realistic setting in which to test different solutions to this problem. We agree and would need to see a much more relevant (to practitioners) demonstration of loss of plasticity and continual backprop - eg in an RL setting. And, relatedly, to convincingly respond to reviewer 2's issue 2). At the same time, we are aware that some avenues may be too resource-intensive, and are open to having a discussion on what kind of experiments would be feasible. Let me know if you wish to discuss this eg in a call

We are glad to hear that everyone finds our work to be of interest and appreciates the thoroughness of our demonstrations of loss of plasticity. In our new experiments, we provide demonstrations of loss of plasticity that are much more relevant to practitioners of both supervised and reinforcement learning:

1. We demonstrate loss of plasticity in deep residual networks in a class-incremental setting. Residual network architectures are much more common in supervised learning applications than the networks in our original manuscript. The class-incremental setting is representative of problems where the network has to accumulate knowledge as more data becomes available continually.
2. We demonstrate the loss of plasticity in standard reinforcement learning problems using standard algorithms and hyperparameters. These results show that loss of plasticity affects even stationary reinforcement learning. The current practice simply hides the issue by prematurely stopping training. In both new experiments, continual backpropagation fully overcomes the loss of plasticity.

- There should be a more thorough discussion of other relevant studies in this area, and discussion and comparisons with other mechanisms to counter loss of plasticity beyond continual backprop, raised by refs. We agree with refs that the thorough exposition of the problem is a clear strength of this paper, and that the latter 'solution' aspect is somewhat weaker. Reviewers would need to be convinced that continual backprop is indeed a realistic, deployable and appropriate solution by responding thoroughly to their concerns, or a much clearer discussion of its limitations and/or comparisons with other continual learning approaches would be needed.

Our new results show that continual backpropagation can be directly combined with the standard deep-learning practice in both supervised and reinforcement learning to overcome the loss of plasticity. We have now added a discussion of new relevant studies in all the appropriate places. We also explicitly mentioned that continual backpropagation does not address the forgetting problem as requested by referee #2. We have also added a preliminary comparison with ReDo in the RL experiments, as requested by Referee #1, and it shows that continual backpropagation performs better than ReDo, while ReDo still loses plasticity. We look forward to hearing back from the referees on further strengthening our solution. But note that the main purpose of our paper is to demonstrate loss of plasticity and not to benchmark all the possible solutions to loss of plasticity.

Response to Referee #1:

My main concern on data used is that all of the tasks considered are quite artificial: they are modifications of standard supervised learning methods, but it is not clear how reflective they are of real-world continuous problems. I understand that there may not be standard benchmarks for continual learning, but it would have been better to focus a bit more on existing problems, rather than customized ones. Indeed, as mentioned by the authors in the paper, reinforcement learning is a problem that can be considered as a continual learning problem, yet the authors still chose to modify an existing environment. The authors mention replay buffers and target networks as mechanisms the RL community has used to become "train-once" tasks, but it seems more natural to me to use RL tasks without a replay buffer and target network, rather than modifying the environment dynamics.

We agree that the problems presented in the original manuscript are somewhat artificial. In the new manuscript, we have added two new experiments. The first one is a class incremental setting, which uses architectures that are much more representative of architectures used in the real world. The second is reinforcement learning experiments in stationary environments. Note that reinforcement learning algorithms have to learn from a non-stationary data stream. However, it is not a fully continual learning problem because once the agent finds the optimal policy, it does not need to learn anything new. However, our results in stationary RL environments show that standard RL methods suffer from loss of plasticity even in stationary environments. Replay buffers and target networks also play other roles in reinforcement learning, like mitigating forgetting. One of our long-term goals is to develop RL methods that learn without buffers or target networks but that require us to address additional issues like forgetting. Overcoming loss of plasticity is an important step towards that goal, but it does not take us all the way to buffer-free RL.

The authors are convincing in demonstrating that existing methods from deep learning are unable to address plasticity loss in continual learning tasks. Although not terribly surprising, as these methods were not developed for continual learning problems, it is still a valid finding. I am a bit less convinced about the robustness of their proposed method (continual backprop). It seems to be quite expensive to run (but neither runtime nor walltime are reported, yet it seems like the advantages over shrink and perturb and the "Mean-corrected contribution" ablation are rather marginal.

We are pleased to know that you find our demonstrations of loss of plasticity in deep learning convincing. We want to clarify that continual backpropagation is not very expensive to run. Perhaps our original manuscript was unclear about this, but in general, continual backprop only reinitializes one feature in each layer every few hundred steps. In our experiments with CIFAR-100, the training time (not the full experiment time, just the clock cycles used while training) for the base deep-learning system was 4.75 hours, while it was 5.6 hours for continual backprop. That is only an 18% overhead. Note that running time can easily be reduced further. The most expensive computational step in continual backpropagation is to keep a running average of utility. However, for cases with mini-batches, like CIFAR-100, we can just use the instantaneous utility. The instantaneous utility is only computed at the time of replacement. Our experiment in an RL environment (A.9d) shows no significant difference in instantaneous utility and a running average of utility. We presented the algorithm in Section A.1 with a running average of utility because it is a general algorithm that can also be used for cases where inputs are shown one at a time, not as mini-batches. Finally, our results on CIFAR-100 show that shrink-and-perturb helps, but it still suffers from loss of plasticity, which continual backprop does not.

1. You mention replay buffers, batching, and target networks make RL "very nearly a train-once setting", but it's not entirely clear why, so it would be good to elaborate this a bit more. For instance, if you add a replay buffer to one of the continual learning tasks you consider, is it no longer a continual learning task?

Replay buffers reduce the effect of new data, making the problem less non-stationary, but it is still continual. We are not using the "train-once" terminology in the revised article.

2. Continuing the above point, it's not clear where RL lies in your definition of continual learning. At one point you say "whenever the house's layout changed... the robot could simply learn from the new data", but you're conflating changing data versus changing environments, which are two different things. A more general term, that you do use further down, is "changes in their data streams". I would suggest you adopt this one everywhere.

Stationary RL is an interesting case. As mentioned earlier, it requires learning from a non-stationary data stream, but only for a while. The agent only has to learn new things until it finds the optimal policy. After that, no new learning is required in stationary RL. Thank you for suggesting using the more general term. We used it in the revised article.

3. You say "A classical deep learning problem like ImageNet is the best place to show the issue of loss of plasticity", but this is not necessarily true. You're using a customized version of ImageNet, not the original benchmark.

We are using the ImageNet dataset, but yes it is not the original benchmark. We have changed the wording where we introduce ImageNet, and it is more precise now.

4. In lines 149-151, couldn't you just output one number, corresponding to the probability of belonging to one of the classes, instead of two heads?

Yes, we could've just outputted one number, which would be equivalent. We preferred to have multiple heads because it made our implementation more general.

5. If adding new heads for new classes is the standard in these types of problems, then I don't think resetting heads, as you do, is equivalent.

Sorry for the confusion. We meant "new heads for new tasks". We have corrected it now.

6. In line 215-216, if you did not use convolutional layers, does this mean the images were flattened? In what order are the pixels presented in this case?

Yes, the images were flattened. MNIST has greyscale images with dimensions 28*28 (784). The order in which the images were flattened is not relevant because we applied a random permutation after flattening.

7. In Figure 3, it would be good to add the values in the non-continual learning setting as reference.

That is a good suggestion. However, to keep the article as small as possible, we decided not to include these baselines. But, if we find a simple way to explain this non-continual baseline, we will add these lines.

8. In Figure 3, right panel, what is the maximum possible value?

We plot the scaled effective rank, which is between 0 and 100. This measure has now been moved to A.7c.

9. In line 378 when you say "improving the condition number", does this mean increasing or decreasing the ratio?

We mean "decreasing the condition number." Thank you for pointing out the possible confusion; we have reflected this change in the paper now.

10. In line 412 you say "we only present results for one set of these hyperparameters". But in the appendix you say "for all the methods except for dropout, we selected the hyperparameter setting that resulted in the highest average percent correct during the whole training period". Why was dropout handled differently?

Dropout performed best when the dropout probability was 0. If we plot dropout with a probability of 0, it'll be the same as backpropagation. But we want to show performance of dropout as used in practice, so we chose dropout with a probability of 0.1. This point is also mentioned in the text.

11. In lines 456-462, is this just poor hparam choice? In Appendix A Adam with stepsize $1e-5$ had better performance. Why didn't you use that? Further, your discussion in lines 466-468 go against the strong results we see with deep RL and Adam, which suggest that the findings might be more due to poor hparam choice.

We found a small bug in our experiment with Adam. We were using beta values slightly different from the standard values. After fixing the bug, Adam's performance improved a little, but it is still the worst of all the algorithms we used. Further, our experiments in RL show that standard Adam is not a good choice in RL, which supports the results from the MNIST experiments.

12. In line 509 when you say "a new hidden unit is added", do you mean reinitialized?

Yes, we meant when a hidden unit is reinitialized. We've edited the text to reflect this.

13. Instead of using a replacement-rate, you could also use a threshold on utility instead (similar to [1]). It would be nice if this were evaluated.

Using a threshold on utility instead of a constant replacement rate is possible. We look forward to future work that compares the two options. But one potential issue with a threshold on utility (the way it was done in [1]) is that the threshold is applied on relative utility, meaning that if the utility of all the units goes down, the threshold-based utility will not replace a sufficient number of units.

14. The math in figure 5 could be made clearer.

We have removed that figure because the new utility measure is simpler, and it is well explained by equation A.1.

15. I appreciate the motivation of your utility measure, but it is quite complicated and it's not clear it needs to be. As mentioned elsewhere in this review, your results with "Mean-corrected contribution" in your ablations, as well as the comparison done in [1] suggest that the utility measure can be much simpler. Perhaps you do need it for your particular setting, but it would require a comparison with ReDo in order to make it convincing.

We found that the utility measure in the original manuscript had a significant advantage over the contribution utility for activation functions that saturate at a non-zero value, like sigmoid, tanh, and ELUs. But for ReLU-type activations, contribution utility performed almost as well as the utility metric in the original submission. So, we chose to only report results with the contribution utility in the new manuscript.

16. On a similar note: it seems very expensive to do the reinitialization at each update. Why not do this less frequently, as was done in [1], for instance?

Continual backpropagation does not reinitialize on each update. For a replacement rate of $1e-6$, in a layer with 500 hidden units, one unit will be replaced after approximately 2000 updates. The reinitialization in continual backprop is quite slow. This may have been unclear in the original manuscript, but Algorithm 1 in the revised manuscript reflects this slow replacement more directly.

17. Given the strong connection between reinforcement learning and continual learning, why not include the reinforcement learning results in the main paper?

They are somewhat hidden in the appendix and there is only a rather small discussion in the main paper.

Thank you for your suggestion. We have moved that reinforcement learning result in the main paper and added some reinforcement learning experiments in stationary environments.

18. In line 670 you say "the benefits of injected noise are controversial because it can be tricky to inject noise to avoid poor performance". Why not try them in your settings? They seem like natural baselines.

Shrink-and-perturb is noise injection with l2 regularization. Our experiments found that noise injection without L2 regularization was not very helpful. We think this is probably because the weights keep increasing over time, reducing the effect of noise.

G (References): This is one of the weakest points in this paper, as the authors are missing proper discussions of some papers directly related to their work. In particular, [1] is exploring a very similar issue (but named dormant neurons) and propose a simpler algorithm for addressing them. In fact, [1] does compare against continual backprop, based on the ArXiv version of this submission, in Appendix C.4. Their results suggest there is no real gain from the scoring method used here, so this requires much further discussion. Some places where I think more comparisons with this paper are necessary are:

1. Lines 288-293
2. Need to compare to ReDo from [1] in section 5 You do cite [1] in the very last paragraph of your submission, and only listed amongst other papers. Given that this is the most related paper, relegating it to the end in such a shallow manner is certainly not enough for publication.

We would like to point out that our original paper on continual backpropagation [2] has been publicly available for a lot longer than ReDo, as also noted by Referee #3. Their work on dormant neurons and Redo is directly inspired by our paper and the talks we gave in Montreal in the Summer and Fall of 2022 [See slide 17 in 3]. The primary point of our paper is to demonstrate the problem of loss of plasticity, not to find the best possible solution to the problem. However, we do realize that it is important to show that continual backpropagation is not significantly worse than any other method. So, we added a comparison with ReDo on the Ant environment, which is the hardest RL environment, and we found that even properly tuned ReDo does not solve performance degradation. Additionally, we do not think that the results in appendix C.4 in the arxiv version of the ReDo paper are representative of continual backprop. The results of the ReDo paper attributed to continual backprop only uses the utility function proposed in the old ArXiv paper with the fixed recycling schedule they proposed in the ReDo paper. Thus, we don't consider it a representative comparison of our algorithm as it is presented in this manuscript or the original arxiv version [2]. We have now cited the ReDo paper when discussing similar methods.

Some other works that should be cited

We have added the references that you pointed out. Note that we can only have upto 50 references in the main paper. So, we had to make choices on which references go in the main paper and which go to Methods.

Thank you for your detailed comments; they have helped us make the article clearer and more relevant to practitioners. We look forward to continuing our discussion with you.

[1] Sokar, G., Agarwal, R., Castro, P.S. & Evci, U.. (2023). The Dormant Neuron Phenomenon in Deep Reinforcement Learning. Proceedings of the 40th International Conference on Machine Learning, in Proceedings of Machine Learning Research 202:32145-32168 Available from <https://proceedings.mlr.press/v202/sokar23a.html>.

[2] Dohare, S., Sutton, R. S., & Mahmood, A. R. (2021). Continual backprop: Stochastic gradient descent with persistent randomness. arXiv preprint arXiv:2108.06325.

[3] Sutton, R. S., & Dohare, S. Maintaining Plasticity in Deep Continual Learning (CoLLAs keynote, Aug 24, 2022) <http://incompleteideas.net/Talks/Talks.html#plasticity>

Response to Referee #2:

This manuscript provides a neat demonstration that deep neural networks sequentially trained on many tasks can suffer from substantial loss of plasticity. While, as the authors themselves discuss in the manuscript, this observation is not completely novel or unexpected, I agree with the authors' claim that they provide the clearest and most direct demonstration so far. Moreover, loss of plasticity is a rather under-appreciated phenomenon in both the reinforcement learning and the continual learning literature, and this manuscript does a great job highlighting this phenomenon and arguing why it is important. In this regard it also greatly helps that the article is skilfully written and accessible to a broad audience. In short, I think that the part of this manuscript that demonstrates the problem of plasticity loss in deep continual learning is generally of high quality and likely to be impactful. The analysis aimed at understanding the factors underlying loss of plasticity is also interesting and nicely done.

However, I was rather disappointed by the solution – continual backpropagation – that is proposed in the manuscript, as well as by the way this solution is evaluated. It seems obvious that periodically reinitialising part of the network should help mitigate loss of plasticity. It also does not seem very striking or interesting that such periodic reinitialising does this more effectively than methods with different motivations such as L2-regularisation, dropout or online normalisation. Instead, if this proposed solution is included, I would expect the authors to address questions such as whether periodically reinitialising part of the network can be done without catastrophic forgetting (i.e., whether this approach strikes a good balance in the plasticity/stability dilemma, e.g. better than current approaches for continual learning), or what the benefit is of only partly reinitialising the network compared to fully reinitialising it. Perhaps an option could also be to leave out the part about the proposed solution, or to include it in minimal form with a clear discussion of its limitations?

We are happy to see that you consider our demonstrations of loss of plasticity as "generally of high quality and likely to be impactful". We want to point out that selective reinitialization, as done by continual backpropagation, is a slow process. For example, in the permuted MNIST experiment, when we use a replacement rate $1e-6$, only one unit is replaced every 500 examples. During a task with 60,000 examples, 120 replacements happen. This means at most 120 (out of 2000) different units are replaced, but typically, it is less than 120 as the same unit can get replaced many times. We believe this replacement is slow enough that continual backpropagation can be combined with methods that mitigate forgetting. With that said, we realize that continual backpropagation is not the paper's main point. Based on your suggestion, we have significantly reduced the emphasis on continual backpropagation. Most of the main body now focuses on the demonstration of loss of plasticity. Additionally, we now explicitly mention in the extended discussion that continual backpropagation does not address forgetting, and a paragraph that discusses full reinitialization.

{1} Regarding the first part of the manuscript on the description of the problem I have one comment, and that is regarding the claim that loss of plasticity is "more fundamental" than catastrophic forgetting (e.g., in the abstract and in the introduction). This is a strong claim, but I could not find a justification for it in the manuscript. I also think that this claim is too strong, and risks raising too high expectations. Let's take reinforcement learning as an example. The issue of forgetting is well-recognized in this field, and addressing it by means of experience replay has resulted in major improvements. Do the authors want to imply that by addressing the issue of loss of plasticity, even larger improvements can be expected?

We do think that plasticity is more fundamental than forgetting. Forgetting is a very important issue for cases with a memory constraint. The memory-constrained setting is very important, but there are other settings. On the other hand, the need for plasticity arises whenever there is new data, even if there is no memory constraint. One example would be continual training of LLMs, where all the old data can be stored, but there is still a need to learn new things as new data arrives. However, we agree that it is not important for the paper to argue what is more fundamental. So, based on your suggestion, we have removed the parts where we were comparing the "fundamentalness" of forgetting and plasticity.

{2} The manuscript is rather naive with respect to the existing continual learning literature, especially when it comes to the evaluation of the proposed continual backpropagation. It is generally agreed in the literature that the goal of continual learning is developing models that can incrementally learn new skills with minimal forgetting of past skills, but this manuscript simply ignores forgetting, and does not evaluate the performance of the continually trained models on previously learned tasks. The justification for this seems to be that the focus of this manuscript is on loss of plasticity, and that this is complementary to the already more-common focus on catastrophic forgetting. While I agree that a *focus* on loss of plasticity is original and commendable (this perspective could help to develop better methods for continual learning), this does not mean that it is interesting or justified to then only evaluate plasticity in isolation from forgetting. This is like studying the plasticity/stability dilemma by only considering plasticity: if stability is disregarded, of course maximising plasticity becomes a lot easier, but also much less interesting. (Note that it is also not the case that current continual learning studies only evaluate forgetting: the main metric in continual learning is average final performance over all tasks, which includes both forgetting and plasticity.)

The online evaluation metric is an appropriate metric for continual learning systems. For any system in the real world, the only thing that matters is how well it does on future data. Performance on old data is only relevant to the extent that past data re-appears in the future. If past data does not reappear, remembering past information is a waste of resources. The online evaluation metric perfectly captures the importance of forgetting. The online performance metric also motivates the community to develop methods that can choose what they remember. A large focus of the current literature is on creating methods that remember everything, but systems that selectively remember previous information can be more efficient. We think that the online performance metric is the ideal metric for continual learning. However, we do realize that continual backpropagation does not address forgetting, and we explicitly mention in the discussion that continual backpropagation does not do so and that an important direction for future work is to develop learning systems that maintain both plasticity and stability. We think that CBP can be easily combined with many methods that address forgetting without making very big changes to the algorithm.

{3} The authors should discuss what is the benefit of only partially re-initialising the network compared to the naive solution of fully re-initialising the network for each new task. I also think that this solution of complete re-initialisation for each new task should be included in the empirical comparisons. For the experiments on ImageNet, my impression is that this naive baseline would be very competitive with continual backpropagation (at least when only considering the performance on the current task). From Figure 1 we can see that the performance for a (re-)initialised network trained on a single task is near 90% (and this could be higher if the step size is tuned for maximal performance after a single task). If a network would be fully re-initialised for each new task, this same performance would be expected for each task in the task sequence. Comparing this with Figure 7, this means that for most tasks in the task sequence the performance of the naive baseline of full re-initialisation would be higher than the performance of continual backpropagation.

We have added a paragraph discussing the full reinitialization solution in the extended discussion section. Full reinitialization has problems in both supervised and reinforcement learning. In supervised learning, it is unable to take advantage of previously learned information. In our Imagenet and CIFAR 100 experiments, we plot the final accuracy, not the time it takes to get to that accuracy. In the online MNIST experiment, the plotted measure takes into account how fast learning happens, and it shows that continual backpropagation is better than full reinitialization (the accuracy of the reinitialized model is the same as the first point in the MNIST plot). We observe the same thing in the ImageNet; retraining takes about 125 epochs to reach peak performance, while continual backdrop gets to peak performance on the 5000th task in just ten epochs. Continual backprop was more than ten times faster than retraining on Continual ImageNet. We have mentioned this point in the discussion, but if you think we should also add a figure showing this, we can do that. Additionally, continual backpropagation can be combined with other methods that mitigate forgetting, which will further improve the computational efficiency of network learning via continual backpropagation. Complete resetting is not practical in the reinforcement learning experiments as the system is learning online. It will simply forget everything it has learned, and its performance will be down to the starting point. Continual backpropagation, compared to full reinitialization, also allows us to train a deployed model without waiting for a new training cycle to finish. This is important when the system learns quickly or the retraining time is too long.

{4} The claim that “the standard practice in deep continual learning [...] is to introduce new heads for new classes” (L161-162, p6) is not quite true. It would be fair to claim that introducing new heads is the standard practice for task-incremental learning, but for other types of continual learning such as domain- and class-incremental learning this is not the case.

Thank you for pointing out this important difference. We have clarified this distinction in the paper. In the new class-incremental CIFAR100 task, we do not reset heads.

Minor things:

- L65 on p24: last mention of supervised should be reinforcement learning?
- L744 on p27: "optimizes" -> "optimizers"

Thank you for pointing out the typos, we have fixed these issues now.

As a final comment, the layout of the submitted manuscript was distorted at several places (e.g., formulas and some figures). For the reviewing I therefore mostly based myself on the latest arXiv version of this manuscript (<https://arxiv.org/abs/2306.13812v2>), which based on spot checks seemed similar to the submitted manuscript except with better layout.

Thank you for accommodating this issue. Yes, the arxiv version you reviewed is the same as the submitted manuscript.

Thank you for your thoughtful feedback; it has helped us to put our paper in a wider context of the continual learning community. We look forward to hearing from you to further strengthen our paper.

Response to Referee #3:

The work is original. In particular the authors are the first to start the discussion around this form of loss of plasticity (through an arxiv paper), and the work spurred a few follow-ups, particularly in the RL community, and therefore it already had an impact.

We are happy to see that you recognize the role of our arxiv paper in the RL community.

I'm not sure I agree with the "train-once" terminology. Traditionally people will refer to this as either the stationary setting or the IID setting. I'm not sure why we need a new term. In my opinion this just creates confusion. Is the train-once meant to convey something different from saying the IID setting / stationary setting? I'm also worried about confusion with one-epoch strategies, where each data point can only be seen once, which have become more popular lately.

Thank you for pointing out the confusion the "train-once" terminology creates. We realize that it is not central to the paper, and we do not use this terminology in the revised manuscript.

Ash & Adams original work on warm-starting, which is also the starting point of Berariu et al., explores a (slightly) different phenomenon than the one described in this manuscript. ... So I'd like the authors to motivate properly why it might be the same phenomenon if that is the stance they would like to take, or to acknowledge that these works might be studying something different.

We agree that these two phenomena are somewhat different. However, we are not sure that they are fundamentally different. In the Imagenet experiment, for the first few hundred tasks, the learning system maintains 100% training accuracy, but the test accuracy drops during these tasks. But as training continues, the training accuracy also starts to drop. We suspect that the loss of generalization ability is just the first phase of the loss of optimization ability. However, we realize that it is unclear at the moment if one of these phenomena is subsumed by the other and if the same underlying cause can explain both. A paragraph in the extended discussion acknowledges the difference between these two phenomena.

I think the authors should better justify the choice of architecture for ImageNet. ... I think it is ok not to run resnet because the target might be RL. But for any computer vision person the choice is unnatural, and acknowledging this point would be useful.

We realize that the architecture used in the ImageNet experiment is not the most popular architecture in computer vision. In our new experiments in class-incremental CIFAR-100, we used an 18-layer ResNet with standard techniques like batch norm, data augmentation, etc. This experiment is a lot closer to the standard practice in computer vision. We think this new experiment addresses your concerns.

All datasets provided are synthetic. In particular permuted MNIST is generally seen as being too toyish in the continual learning community and people are dissuaded from using it. I can understand that here the authors go through 150 tasks (vs typical Continual learning papers that would do 10). But an issue with permuted MNIST is not scale but the fact that there is no structure to share between the tasks. It would be interesting to know what happens when there is common structure between tasks as one would expect in more realistic scenarios (ImageNet represents this scenario). In that sense I would suggest to bring the ant task from the appendix into the main text, since it is in my view more interesting than the permuted MNIST.

Indeed, permuted MNIST does not have a shared structure across tasks. We think this makes permuted MNIST a good benchmark for studying plasticity because it has minimal forward transfer, which can be a confounder when we explicitly try to study plasticity. However, we agree it is not a very realistic problem, and we have moved it to the appendix. Thank you for your suggestion of bringing the Ant task to the main paper; we've done so. We have also added experiments from stationary RL tasks and class-incremental CIFAR-100 to the main paper. These tasks are much more realistic than Permuted MNIST.

Comment on dropout. Since it encourages redundant representation (i.e. same information being encoded in multiple features) wouldn't one expect to reduce the rank and hence not help? I would have preferred a more technical explanation of how each method should help. E.g. for Adam, why should it help with non-stationarity? The algorithm was developed for the IID setting and aims to make better use of curvature information. Has anyone connected its use in RL with non-stationarity? I could see it as just being a better optimizer but as affected by non-stationarity as SGD. I agree that it is important to run experiments with different optimizers, particularly widely used ones like Adam. But the motivation is different. Not that they should help, but rather just to make sure the results hold in typical scenarios. From how the text is written, it implies that there are some reasons we should expect Adam to help improve plasticity (which is actually not spelled out or explored).

To reduce the size of the paper, we decided to remove the speculative part about why different methods will help with plasticity loss. We now directly present the results after introducing the method.

The authors identify 3 causes for loss of plasticity: dead units, weight magnitude and low rank. The authors do not fully go into explaining why these 3 affect plasticity. E.g. for dead units the paper states that later on (next section) will show that if we reduce the number of dead units we have more plasticity, implying their causal role in plasticity loss. Fair enough, but it would also help to explain how they are connected. E.g. say something to the effect that dead units do not allow gradients to flow, hence you can not change weights going in the dead unit. For weight magnitude authors mention exploding gradients in RNNs, which is potentially irrelevant for the work since there are just shallow feedforward networks. A more relevant approach was to connect this to curvature and potentially ill-conditioning of the optimizer. Basically rely on traditional tools to understand learning and being more precise. The gradients being bounded argument also doesn't hold, because they are technically bounded when learning fails (probably larger than those on the early tasks, but bounded). Right now there is very little precise information of why these 3 metrics matter and no discussion whether there are other properties that matter. And the authors do not really provide a technical description of how these three items interfere with the learning process, leaving things in my opinion a bit too vague.

Based on your suggestion, we have provided some more technical details on how dead units and large weights contribute to plasticity loss. Let us know if there are some more technical details that you'd like us to add.

In the same spirit of the comment above. When introducing the utility measure, could the authors provide more precise technical reasons of why the metric makes sense? E.g. the authors mention sparsity using a similar metric. Usually in sparsity the metric is motivated using Taylor expansions that gives you a mathematical tool to say how much zeroing out a weight will affect the output of the function. Similarly, adaptability maybe can be connected with how much the function can change under an update step for some metric? None of this is done in the paper. The explanation is intuitive, but not very technical or precise.

It is indeed a limitation of our current utility measure that it is based on a heuristic. Using a Taylor expansion to measure how much zeroing out a unit will affect the output will give us a different utility measure and a slightly different ranking of units. We look forward to future work that brings these utility metrics to plasticity research.

Subtracting the average of utility might imply a Gaussian assumption on the activation, right? Such assumptions should be made explicit. What other assumptions are made in terms of how activations are meant to work?

We found that in our new experiments, subtracting the average of utility did not significantly improve performance. So, we chose to remove this step from our algorithm to make the algorithm more easily adoptable by the community.

Connected to a previous point about permuted MNIST, but I think important for the paper is a discussion on the tradeoffs between maintaining ability to learn, being able to learn to reuse previous knowledge, performance that the model achieves on the given task, maybe even catastrophic forgetting. In particular I find that the authors do not integrate the work into the different concerns that people have in the community. I would appreciate a paragraph describing these different phenomena that have been studied in the community. For example, a considerable section of the community is focusing on the concept of forward transfer. The goal is not only to be able to keep learning, but also to learn more efficiently as time goes by because you get to reuse skills and knowledge that you have accumulated so far. From this perspective, fixing the optimization to have the neural network able to still learn feels like a first step but not sufficient. If future data becomes similar (but not the same) with previous data, then resetting certain units that seem to have low utility might be harmful. These considerations affect the choice of benchmark. For example, in permuted MNIST there is no shared structure, so questions like whether continual backprop learns as fast (or faster) than learning from scratch can not be asked in a meaningful way. ImageNet is not potentially set up in such a way to see whether one can find itself in a situation where locally a unit might seem not useful, but globally it will be. I'm not suggesting that the authors need to address or experiment with any of these alternatives. But I think acknowledging the existence of these themes, and how the choice of benchmark affect these questions will give to the reader a much better and rounded perspective of the topics explored in the community and the role of loss of plasticity within this context.

Thank you for this comment, we appreciate the importance of placing plasticity and the benchmarks we used in a wider context. We have added a paragraph in the discussion that addresses this issue. Please let us know if you think there are other aspects of continual learning, or their relationship to plasticity should be highlighted in our paper.

Thank you for your comments, they have improved our article. We look forward to having further discussion with you.

Reviewer Reports on the First Revision:

Referees' comments:

Referee #1 (Remarks to the Author):

A. Summary of the key results

This paper focuses on continual learning with deep networks and argues that most (if not all) existing methods are unable to deal with continual learning setups. The authors argue that this failure is due to loss of plasticity in the neural network. They demonstrate this behaviour in a few supervised and reinforcement learning tasks and methods. Finally, the authors propose continual backpropagation as a new algorithm for mitigating loss of plasticity.

=====

B. Originality and significance: if not novel, please include reference

Plasticity loss is something that has already been receiving a fair bit of attention, as referenced in the paper itself [11, 12, 16, 17, 18, 19, 20, 21, 22, 23, 24, 25, 26]. What is novel in this work is the examination of this plasticity loss in continual learning settings. The continual backprop algorithm suggested is also novel, appearing only in the preprint by the same authors [21].

Plasticity loss is in general an important topic in deep learning.

=====

C. Data & methodology: validity of approach, quality of data, quality of presentation

Although I appreciate the effort the authors put into addressing the concerns in the first round of reviews, I still feel the paper is not sufficiently convincing for the claims being made. This is a combination of some of the motivating claims made at the beginning of the paper, as well as some of the experimental design decisions made:

1. In the preamble, the authors say "Deep learning is usually restricted to a special training phase and then turned off when the network is actually used... in the second phase, continued learning is not possible because there are no labels... The problem is not well understood, and there is not as yet a clear solution... the most effective strategy for incorporating new data has been simply to discard the old network and train a new one from scratch on the old and new data together." In the discussion they say "Deep learning is an effective and valuable technology in settings where learning occurs once in a special training phase and never thereafter."

These statements are simply not true, as supervised fine-tuning (SFT) is doing exactly this: continue training a network on new labels. This has likely become the most popular way of adapting large language models (LLMs) to specific tasks, either via an existing dataset or through RLHF. If continued training were truly not possible, none of these methods would work, which is clearly not the case.

Even in the reinforcement learning setting there has been prior work demonstrating that one can "reincarnate" past models/policies without having to retrain from scratch (Agarwal et al., 2022, <https://arxiv.org/abs/2206.01626>).

2. "In most of our results we use an online performance measure, the error on each example when presented, with each example presented only once."

Why is each example presented only once? This seems unnecessary and not really reflective of how things would be done in practice, where we might store new data in memory to be able to retrain on it more than once.

3. Section 1 starts with "The primary purpose of this article is to demonstrate loss of plasticity in standard deep learning systems." Given that this is the driving force of the paper, it would be nice for "loss of plasticity" to be defined a bit more formally or precisely. Figure 2 is doing some of this work, but unfortunately the "stable rank" presented in the right panel is not formally defined (only an intuition is provided in the image caption).

4. In describing Continual ImageNet the authors say: "After training and testing on one task, the next task began with a different pair of classes." Is this abrupt change really reflective of "natural learning", as claimed in the abstract? Again, it doesn't seem like it properly captures realistic training scenarios.

5. In Figure 1 a horizontal green line represents a "Linear Network", but no details are provided. Is this linear network trained on all tasks? Separately for each task? Why doesn't the accuracy change with different task numbers?

6. In discussing Figure 1, the authors say "We found this to be a common pattern in our experiments: for a well-tuned network, performance first improves, then falls substantially, ending near or below the linear baseline." People often use learning rate schedules to deal exactly with this type of scenario. Wouldn't learning rate schedules help deal with this? Perhaps not, but it would be more convincing if this were explored, as this is something commonly used in practice.

7. The authors claim that the results from Figure 1 are "substantial evidence that deep learning methods do not work well in continual learning problems." This is a strong statement that I don't feel is properly supported by the provided evidence (see points 4, 5, and 6 above).

8. In discussing the setup for Figure 2, the authors say "We also used additional deep-learning techniques, including batch normalization, data augmentation, L2 regularization, and learning rate scheduling." Why weren't these used in the Continual ImageNet experiments? This would be especially important given point 7 above.

9. In the end of the first paragraph in section 3, the authors say their results "suggest that small weights and sustained variability in the network may be important for maintaining plasticity." However, in the previous paragraph the authors say "The addition of L2 regularization ... resulted in very small weights, which prevent the agent from committing to good behaviour." These two statements seem to be somewhat contradictory, and it's also not clear why "very small weights"

would result in an inability to "commit to good behaviour."

10. In the discussion, the authors say "Taking a closer look, we found that during training many of the networks' neuron-like units become ... similar to each other." Where is this measured? Similar according to what metric? This needs to be explained, as it is used as evidence for the "loss of plasticity" being argued throughout.

=====

D. Appropriate use of statistics and treatment of uncertainties

The results presented seem appropriate with means and standard deviations presented over a number of independent runs.

=====

E. Conclusions: robustness, validity, reliability

As discussed in detail in section C above, I don't feel the paper is sufficiently convincing for the claims being made.

=====

F. Suggested improvements: experiments, data for possible revision

I would recommend the authors soften the claims made and try to focus on settings which are closer to what would be used in practice (e.g. I don't feel Continual ImageNet is very reflective of realistic training scenarios). There are some very interesting insights provided here, but the methods investigated are not comprehensive enough to justify the general claims being made.

=====

G. References: appropriate credit to previous work?

I found the references to prior work to be appropriate.

=====

H. Clarity and context: lucidity of abstract/summary, appropriateness of abstract, introduction and conclusions

As mentioned above, I feel the claims made in the abstract/intro and conclusions are too strong given the provided evidence.

Referee #1 (Remarks on code availability):

I did a quick examination of the code and it seems to be cleanly written and including the implementations used in the submission.

Referee #2 (Remarks to the Author):

Thanks to the authors for a detailed response to my review. I am mostly satisfied with the answers in the rebuttal and the changes to the manuscript. However, I still have a couple of comments and concerns.

Most of my concerns relate to L48-L52 in the updated manuscript: “Loss of plasticity is different from catastrophic forgetting, which concerns poor performance on old examples that are not re-presented [27, 28, 29]. In most of our results we use an online performance measure, the error on each example when presented, with each example presented only once. This performance measure includes evaluation of forgetting only to the extent that forgetting about earlier examples impairs performance on future examples.”

Firstly, it is a mischaracterization that catastrophic forgetting is about poor performance on old examples that are not re-presented. Why do the authors add “that are not re-presented” here? Catastrophic forgetting is about the performance on any old examples, regardless of whether or not they are re-presented.

Then I think there are a number of issues with the “online performance measure”.

(1) In the rebuttal, the authors claim that “the online performance metric is the ideal metric for continual learning” and that “the online evaluation metric perfectly capture the importance of forgetting”, because “performance on old data is only relevant to the extent that past data re-appears in the future”. I could agree with this argument if one would deal with real-world or natural data streams in which there is a rich, realistic temporal correlation structure. In that case it could be claimed that the online performance metric captures the ‘right’ trade-off between forgetting and plasticity. However, this is not the case in this paper. This paper uses artificial data streams in which old tasks are never repeated. Because of this, the online performance metric is heavily, if not fully, biased towards plasticity. When used in this kind of data streams, this metric does not adequately capture the importance that forgetting would play in more natural or realistic settings. (See also below after “Thirdly, ...”.)

(2) The term “online performance measure” is not fully appropriate for this paper, because for the experiments on both ImageNet and CIFAR-100, the evaluation is not really “online” (and not as described in L50-51). In both cases, the networks are first extensively trained on the new task (for 200-250 epochs), before the performance on them is evaluated. For an “online” measure, one would expect performance to be evaluated whenever a new training sample arrives.

Thirdly, I think the authors should make it clear that, in the setups they use, the “extent that forgetting about earlier examples impairs performance on future examples” is likely very small. This can for example be concluded from that performance of full reinitialization (which has complete forgetting) is still very good. This means that at least the performance metric that is plotted in Fig 1 and 2 does not seem to capture much, if any, of the effects of forgetting. (It seems to be the case that forgetting does, to some extent, impact the speed with which future examples can be learned, but this is not captured by the metrics in Fig 1 and 2.)

It might be good to clarify that I do think that the currently used data streams and metric are

appropriate for the main points that the authors make (i.e., regarding loss of plasticity and that this can be rescued by injecting relatively small amounts of randomness). I only ask that the authors are clearer and more fair, especially in the main text, in their discussion of forgetting and acknowledge that this is an aspect that is mostly ignored in their evaluations and something that is left for future work. I think this is especially important given that the authors describe their experiments as “continual learning”, while in the current continual learning literature forgetting does typically play an important role. (Arguably, perhaps a too large role, but that is a separate discussion.)

Finally, for the class-incremental learning experiments on CIFAR-100, I think it is important that the authors explain already in the main text that their setup differs from the typical class incremental setting (and from the one used in ref [38]) by allowing access to the training data of all classes encountered so far. Again, I think this setting is appropriate to use in this paper, but to avoid confusion I think it is important to point out this difference straight away. (Especially as currently in the main text the class-incremental learning setting is introduced by referring to ref [38], but without clarifying this important difference.) Also, in L432, I think the authors should leave out “small”, as I do not think this is a “small deviation”.

Comment regarding statistics:

- It seems that for Figures 2, 3, 4, A.6 and A.9 it is not described how many runs the plots are based on, and the shaded regions are not defined. For the other figures the descriptions do seem appropriate.

Minor thing:

- It seems L467-468 (“Finally, for the head resetting baseline ...”) describes a baseline that is not included in the presented results.

Referee #3 (Remarks to the Author):

The authors describe a form of loss of plasticity for neural networks, and provide a solution, namely continual backprop. The work was first published on arxiv, and generated a great amount of interest within the community (particularly the Continual RL community). Therefore I think it is worth recognizing the authors for the impact they had.

From my point of view, the work is novel, and has already shown its impact. The paper is written well, and the authors in the revised version addressed multiple of my concerns in terms of framing. The things that are standing, are more a disagreement of opinion, but not necessarily wrong. More descriptive of the authors' point of view. I'm particularly referring to the distinction between inability to optimize vs lack of generalization as two distinct types of loss of plasticity. Personally I believe they are unrelated and should be studied separately, but I concur that this is a question that the community will settle in the near future, and is not up for this work to provide that answer. And the authors have a right to highlight their view (that these things are related). Anyway this does not affect the correctness of their claims.

Overall, taking into account the rebuttal and changes done with the paper (the addition of new experiments) I'm happy with the overall paper and I think it is worth accepting.

Author Rebuttals to First Revision:

Response from the authors

We are glad that all the referees and editors appreciate the new results and the effort we put into the last round of revisions. We thank all the reviewers and referees for their detailed feedback, which has helped us improve the article. In the revised version, we have made minor changes to writing to address the remaining points raised by the referees.

Response to the editor

“ ... It needs to be made clear in which situations this phenomenon applies, and in which it may not (or has not been observed, at least) ... ”

Our results show that loss of plasticity is a critical factor when learning continues for many tasks, but it may be less important if learning happens for a small number of tasks. This can be seen in our CIFAR-100 results, Figure 2, where previous learning helped until the first 40 classes, but loss of plasticity became a critical factor by 100 classes. In some deep learning applications, the learning system is first trained on a large dataset and then fine-tuned on a smaller dataset, and performance improves by training on the first dataset. In these applications, there are only two tasks. Our results are consistent with the results in these applications because in our results there was an advantage due to previous learning in the first few tasks. We do not expect loss of plasticity to be a major factor in applications with very few tasks. We have added a line at the end of the first paragraph in the discussion to clarify this and added a paragraph in the extended discussion to elaborate on when loss of plasticity becomes a significant issue.

Response to Referee #1

“ ... These statements are simply not true, as supervised fine-tuning (SFT) is doing exactly this: continue training a network on new labels. This has likely become the most popular way of adapting large language models (LLMs) to specific tasks, either via an existing dataset or through RLHF. If continued training were truly not possible, none of these methods would work, which is clearly not the case. Even in the reinforcement learning setting there has been prior work demonstrating that one can "reincarnate" past models/policies without having to retrain from scratch (Agarwal et al., 2022, <https://arxiv.org/abs/2206.01626>). ”

Thank you for pointing it out. Here, we intended to indicate that typically learning is turned off when a network is used, not that learning can not be done. We have updated the statements that you pointed out to reflect that. We have also updated the second paragraph of the introduction to clarify the points about fine-tuning, and we have softened the claims you mentioned in that paragraph and the conclusion. We have also added a paragraph in the extended discussion clarifying that loss of plasticity has a significant effect when learning continues for a large number of tasks, but this effect may not be present when there are only a few tasks. The cases of

supervised fine-tuning that you pointed out only have two tasks, so loss of plasticity may not be significant in these cases.

"Why is each example presented only once? This seems unnecessary and not really reflective of how things would be done in practice, where we might store new data in memory to be able to retrain on it more than once."

We have removed the sentence about examples being presented only once from the introduction. This sentence was not entirely accurate for the experiments in the main paper. For Imagenet, examples from the current classes are stored and used to train the network using mini-batches, and in class-incremental CIFAR100, we store all examples and use them for learning.

"... it would be nice for "loss of plasticity" to be defined a bit more formally or precisely. Figure 2 is doing some of this work, but unfortunately the "stable rank" presented in the right panel is not formally defined (only an intuition is provided in the image caption)."

We have defined loss of plasticity in the first line of the third paragraph of the introduction. We say systems lose plasticity when they "lose their ability to learn." We have also clearly described what it would mean for a system to lose plasticity for both supervised learning experiments. The mathematical definition of stable rank is presented in the Methods section, which details the experimental results in Figure 2.

In describing Continual ImageNet the authors say: "After training and testing on one task, the next task began with a different pair of classes." Is this abrupt change really reflective of "natural learning", as claimed in the abstract? Again, it doesn't seem like it properly captures realistic training scenarios

This experiment may not be reflective of real-world scenarios. However, the point of this experiment is to be scientifically correct and not contain any confounders. The other experiments in the paper are much more focused on natural settings, like the Ant experiment, and applications, like class-incremental CIFAR-100, where the learning system has to accumulate new knowledge over time.

Figure 1 a horizontal green line represents a "Linear Network", but no details are provided. Is this linear network trained on all tasks? Separately for each task? Why doesn't the accuracy change with different task numbers?

In this problem, we reset the head of the network at the beginning of each task. It means that for a linear network, the whole network is reset. That is why the performance of a linear network will stay the same in this problem. As the linear network is a baseline, having a low-variance estimate of its performance is desirable. We achieve this by averaging the performance of a linear network over a large number of tasks. The linear network is trained on many more tasks than other networks in the experiments. This gives us a much better estimate of its performance than other networks. We have added these details to the Methods section as well.

In discussing Figure 1, the authors say "We found this to be a common pattern in our experiments: for a well-tuned network, performance first improves, then falls substantially, ending near or below the linear baseline." People often use learning rate schedules to deal exactly with this type of scenario. Wouldn't learning rate schedules help deal with this? Perhaps not, but it would be more convincing if this were explored, as this is something commonly used in practice.

We explore the use of learning rate scheduling in the CIFAR-100 experiment, and it shows that learning rate scheduling does not fix loss of plasticity. The lack of a learning rate scheduling in this experiment increases the diversity of experiments where we demonstrate loss of plasticity. More generally, scheduling the learning rate to reduce over time is not a realistic solution for continual learning, as the learner is expected to learn indefinitely.

The authors claim that the results from Figure 1 are "substantial evidence that deep learning methods do not work well in continual learning problems." This is a strong statement that I don't feel is properly supported by the provided evidence (see points 4, 5, and 6 above).

We have addressed points 4,5 and 6 above. We agree that this sentence can be better phrased. We have edited it to say that "... is direct evidence that these methods do not work well in continual learning problems," which is accurate. More generally, our experiments all together constitute extensive and substantial evidence of loss of plasticity. We have added a line at the end of section 1 to reflect this conclusion. "Altogether, these results, along with other extensive results in Methods, constitute substantial evidence of plasticity loss."

In discussing the setup for Figure 2, the authors say "We also used additional deep-learning techniques, including batch normalization, data augmentation, L2 regularization, and learning rate scheduling." Why weren't these used in the Continual ImageNet experiments? This would be especially important given point 7 above.

We did not use these techniques in the ImageNet experiment because they would add additional confounders, and we want to keep this experiment clean and without confounders. Additionally, the results in the CIFAR100 experiments show that these techniques do not fix plasticity loss.

In the end of the first paragraph in section 3, the authors say their results "suggest that small weights and sustained variability in the network may be important for maintaining plasticity." However, in the previous paragraph the authors say "The addition of L2 regularization ... resulted in very small weights, which prevent the agent from committing to good behaviour." These two statements seem to be somewhat contradictory, and it's also not clear why "very small weights" would result in an inability to "commit to good behaviour."

Thank you for pointing out this confusion. We meant that non-growing weights seem to be important for maintaining plasticity. We have updated the first line to say that "... suggest that non-growing weights and sustained variability in the network may be important for maintaining plasticity."

In the discussion, the authors say "Taking a closer look, we found that during training many of the networks' neuron-like units become ... similar to each other." Where is this measured? Similar according to what metric? This needs to be explained, as it is used as evidence for the "loss of plasticity" being argued throughout.

Both stable rank and dead units provide a measure of loss of diversity. Stable rank is a direct measure of the diversity of neurons, while all dead units are the same, which means that they are also not diverse.

I would recommend the authors soften the claims made and try to focus on settings which are closer to what would be used in practice (e.g. I don't feel Continual ImageNet is very reflective of realistic training scenarios). There are some very interesting insights provided here, but the methods investigated are not comprehensive enough to justify the general claims being made.

We are glad you find the work provides very interesting insights. Per your suggestions, we have softened the claims that you pointed out in the introduction and conclusion. We have addressed all the points you raised and clarified the text where necessary.

Thank you once again for your feedback. It has strengthened our work and made it more precise.

Response to Referee #2

Thanks to the authors for a detailed response to my review. I am mostly satisfied with the answers in the rebuttal and the changes to the manuscript. However, I still have a couple of comments and concerns.

We are glad to hear that our comments from the previous round clarified most of your concerns. Thank you for your detailed comments from the previous round, they have helped us significantly improve the presentation of our paper.

Most of my concerns relate to L48-L52 in the updated manuscript: “Loss of plasticity is different from catastrophic forgetting, which concerns poor performance on old examples that are not re-presented [27, 28, 29]. In most of our results we use an online performance measure, the error on each example when presented, with each example presented only once. This performance measure includes evaluation of forgetting only to the extent that forgetting about earlier examples impairs performance on future examples.”

Firstly, it is a mischaracterization that catastrophic forgetting is about poor performance on old examples that are not re-presented. Why do the authors add “that are not re-presented” here? Catastrophic forgetting is about the performance on any old examples, regardless of whether or not they are re-presented.

Thank you for raising this point. We agree that our phrasing was not fully accurate. We have edited this sentence to say “... catastrophic forgetting, which concerns poor performance on old examples even if they are not re-presented”. The new phrasing is correct, highlighting the point that forgetting may not be a relevant issue when old data is not presented again.

“Then I think there are a number of issues with the “online performance measure. ...”

We agree that the online performance measure is not fully appropriate for ImageNet and CIFAR-100 experiments, as we used test and training sets in those experiments. We also realize that online performance measure are not critical to the main message of loss of plasticity. We have removed the two sentences about the online performance measure from the introduction. The points made in the main paper do not depend on the online performance measure. We believe that this alleviates all of your concerns about the online performance measure.

Thirdly, I think the authors should make it clear that, in the setups they use, the “extent that forgetting about earlier examples impairs performance on future examples” is likely very small. This can for example be concluded from that performance of full reinitialization (which has complete forgetting) is still very good. This means that at least the performance metric that is plotted in Fig 1 and 2 does not seem to capture much, if any, of the effects of forgetting. (It seems to be the case that forgetting does, to some extent, impact the speed with which future examples can be learned, but this is not captured by the metrics in Fig 1 and 2.)

It might be good to clarify that I do think that the currently used data streams and metric are appropriate for the main points that the authors make (i.e., regarding loss of plasticity and

that this can be rescued by injecting relatively small amounts of randomness). I only ask that the authors are clearer and more fair, especially in the main text, in their discussion of forgetting and acknowledge that this is an aspect that is mostly ignored in their evaluations and something that is left for future work. I think this is especially important given that the authors describe their experiments as “continual learning”, while in the current continual learning literature forgetting does typically play an important role. (Arguably, perhaps a too large role, but that is a separate discussion.)

We agree that the current metrics are appropriate for studying plasticity but not forgetting. We agree that the settings we studied do not focus on forgetting. We have now explicitly mentioned in the extended discussion that our experiments and evaluation metrics are focused on plasticity, not forgetting or other aspects of continual learning.

Finally, for the class-incremental learning experiments on CIFAR-100 ... Also, in L432, I think the authors should leave out “small”, as I do not think this is a “small deviation”.

Thank you for raising this important distinction. In the new description we do not mention that this is a “small deviation”.

It seems that for Figures 2, 3, 4, A.6 and A.9 it is not described how many runs the plots are based on, and the shaded regions are not defined. For the other figures the descriptions do seem appropriate.

Thank you for pointing it out. We have now described the details about runs and shaded regions in Methods and in the Extended Data Figure’s captions.

It seems L467-468 (“Finally, for the head resetting baseline ...”) describes a baseline that is not included in the presented results.

This baseline is not present in the main paper, but it is present in Extended Data Figure 1a. We’ve added that reference when we mention the baseline to ensure that there is no more confusion.

Thank you again for providing valuable feedback. Your feedback has made our work more precise and more relevant to the community. We appreciate the time and effort you have put into your reviews.

Response to Referee #3

The authors describe a form of loss of plasticity for neural networks, and provide a solution, namely continual backprop. The work was first published on arxiv, and generated a great amount of interest within the community (particularly the Continual RL community).

Therefore I think it is worth recognizing the authors for the impact they had.

Thank you. We are glad to hear that you recognize the impact our work has had on the reinforcement learning community.

From my point of view, the work is novel, and has already shown its impact. The paper is written well, and the authors in the revised version addressed multiple of my concerns in terms of framing. The things that are standing, are more a disagreement of opinion, but not necessarily wrong. More descriptive of the authors' point of view. I'm particularly referring to the distinction between inability to optimize vs lack of generalization as two distinct types of loss of plasticity. Personally I believe they are unrelated and should be studied separately, but I concur that this is a question that the community will settle in the near future, and is not up for this work to provide that answer. And the authors have a right to highlight their view (that these things are related). Anyway this does not affect the correctness of their claims.

We agree that this is an issue of difference of opinion and does not affect the correctness of our claims. We are also looking forward to future work that settles these differences.

Overall, taking into account the rebuttal and changes done with the paper (the addition of new experiments) I'm happy with the overall paper and I think it is worth accepting.

Thank you for your comments. They helped us significantly improve the paper and make it more relevant to practitioners and a general audience.